# SDEdit: Guided Image Synthesis and Editing with Stochastic Differential Equations

**Chenlin Meng**[1]   **Yutong He**[1]   **Yang Song**[1]   **Jiaming Song**[1]
**Jiajun Wu**[1]   **Jun-Yan Zhu**[2]   **Stefano Ermon**[1]
[1]Stanford University   [2]Carnegie Mellon University

## Abstract

Guided image synthesis enables everyday users to create and edit photo-realistic images with minimum effort. The key challenge is balancing *faithfulness* to the user inputs (*e.g.*, hand-drawn colored strokes) and *realism* of the synthesized images. Existing GAN-based methods attempt to achieve such balance using either conditional GANs or GAN inversions, which are challenging and often require additional training data or loss functions for individual applications. To address these issues, we introduce a new image synthesis and editing method, Stochastic Differential Editing (SDEdit), based on a diffusion model generative prior, which synthesizes realistic images by iteratively denoising through a stochastic differential equation (SDE). Given an input image with user guide in a form of manipulating RGB pixels, SDEdit first adds noise to the input, then subsequently denoises the resulting image through the SDE prior to increase its realism. SDEdit does not require task-specific training or inversions and can naturally achieve the balance between realism and faithfulness. SDEdit outperforms state-of-the-art GAN-based methods by up to $98.09\%$ on realism and $91.72\%$ on overall satisfaction scores, according to a human perception study, on multiple tasks, including stroke-based image synthesis and editing as well as image compositing.

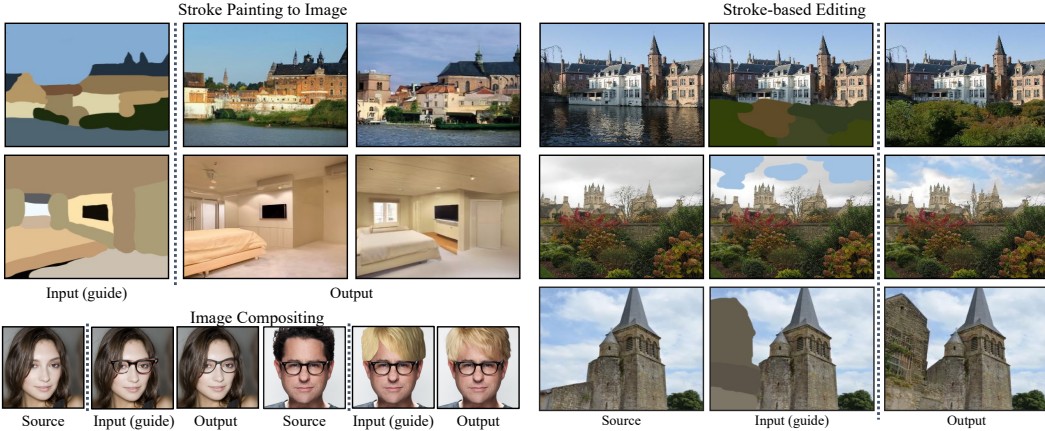

Figure 1: Stochastic Differential Editing (SDEdit) is a **unified** image synthesis and editing framework based on stochastic differential equations. SDEdit allows stroke painting to image, image compositing, and stroke-based editing **without** task-specific model training and loss functions.

## 1 Introduction

Modern generative models can create photo-realistic images from random noise ([Karras et al., 2019](#); [Song et al., 2021](#)), serving as an important tool for visual content creation. Of particular interest is guided image synthesis and editing, where a user specifies a general guide (such as coarse colored strokes) and the generative model learns to fill in the details (see Fig. [1](#)). There are two natural

desiderata for guided image synthesis: the synthesized image should appear *realistic* as well as be *faithful* to the user-guided input, thus enabling people with or without artistic expertise to produce photo-realistic images from different levels of details.

Existing methods often attempt to achieve such balance via two approaches. The first category leverages conditional GANs (Isola et al., 2017; Zhu et al., 2017), which learn a direct mapping from original images to edited ones. Unfortunately, for each new editing task, these methods require data collection and model re-training, both of which could be expensive and time-consuming. The second category leverages GAN inversions (Zhu et al., 2016; Brock et al., 2017; Abdal et al., 2019; Gu et al., 2020; Wu et al., 2021; Abdal et al., 2020), where a pre-trained GAN is used to invert an input image to a latent representation, which is subsequently modified to generate the edited image. This procedure involves manually designing loss functions and optimization procedures for different image editing tasks. Besides, it may sometimes fail to find a latent code that faithfully represents the input (Bau et al., 2019b).

To balance *realism* and *faithfulness* while avoiding the previously mentioned challenges, we introduce SDEdit, a guided image synthesis and editing framework leveraging generative stochastic differential equations (SDEs; Song et al., 2021). Similar to the closely related diffusion models (Sohl-Dickstein et al., 2015; Ho et al., 2020), SDE-based generative models smoothly convert an initial Gaussian noise vector to a realistic image sample through iterative denoising, and have achieved unconditional image synthesis performance comparable to or better than that of GANs (Dhariwal & Nichol, 2021). The key intuition of SDEdit is to "hijack" the generative process of SDE-based generative models, as illustrated in Fig. 2. Given an input image with user guidance input, such as a stroke painting or an image with stroke edits, we can add a suitable amount of noise to smooth out undesirable artifacts and distortions (*e.g.*, unnatural details at stroke pixels), while still preserving the overall structure of the input user guide. We then initialize the SDE with this noisy input, and progressively remove the noise to obtain a denoised result that is both realistic and faithful to the user guidance input (see Fig. 2).

Unlike conditional GANs, SDEdit does not require collecting training images or user annotations for each new task; unlike GAN inversions, SDEdit does not require the design of additional training or task-specific loss functions. SDEdit only uses a single pretrained SDE-based generative model trained on unlabeled data: given a user guide in a form of manipulating RGB pixels, SDEdit adds Gaussian noise to the guide and then run the reverse SDE to synthesize images. SDEdit naturally finds a trade-off between realism and faithfulness: when we add more Gaussian noise and run the SDE for longer, the synthesized images are more realistic but less faithful. We can use this observation to find the right balance between realism and faithfulness.

We demonstrate SDEdit on three applications: stroke-based image synthesis, stroke-based image editing, and image compositing. We show that SDEdit can produce *realistic* and *faithful* images from guides with various levels of fidelity. On stroke-based image synthesis experiments, SDEdit outperforms state-of-the-art GAN-based approaches by up to $98.09\%$ on realism score and $91.72\%$ on overall satisfaction score (measuring both realism and faithfulness) according to human judgements. On image compositing experiments, SDEdit achieves a better faithfulness score and outperforms the baselines by up to $83.73\%$ on overall satisfaction score in user studies. Our code and models will be available upon publication.

## 2 Background: Image Synthesis with Stochastic Differential Equations (SDEs)

Stochastic differential equations (SDEs) generalize ordinary differential equations (ODEs) by injecting random noise into the dynamics. The solution of an SDE is a time-varying random variable (*i.e.*, stochastic process), which we denote as $\mathbf{x}(t) \in \mathbb{R}^d$, where $t \in [0, 1]$ indexes time. In image synthesis (Song et al., 2021), we suppose that $\mathbf{x}(0) \sim p_0 = p_{\text{data}}$ represents a sample from the data distribution and that a forward SDE produces $\mathbf{x}(t)$ for $t \in (0, 1]$ via a Gaussian diffusion. Given $\mathbf{x}(0)$, $\mathbf{x}(t)$ is distributed as a Gaussian distribution:

$$\mathbf{x}(t) = \alpha(t)\mathbf{x}(0) + \sigma(t)\mathbf{z}, \quad \mathbf{z} \sim \mathcal{N}(\mathbf{0}, \boldsymbol{I}), \tag{1}$$

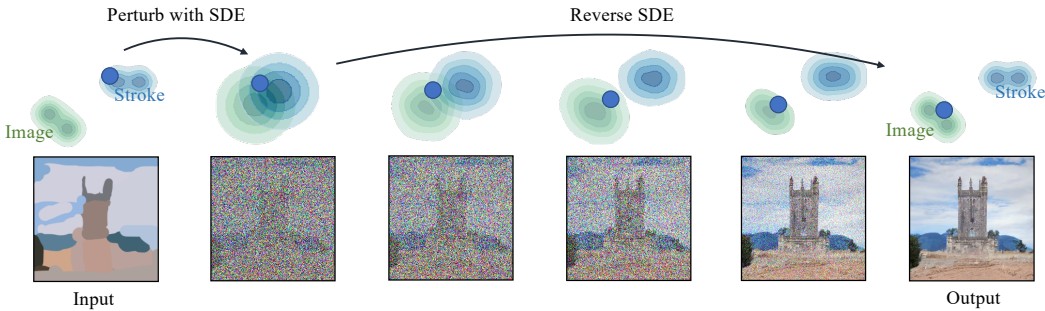

Figure 2: Synthesizing images from strokes with SDEdit. The blue dots illustrate the editing process of our method. The green and blue contour plots represent the distributions of images and stroke paintings, respectively. Given a stroke painting, we first perturb it with Gaussian noise and progressively remove the noise by simulating the reverse SDE. This process gradually projects an unrealistic stroke painting to the manifold of natural images.

where $\sigma(t) : [0, 1] \rightarrow [0, \infty)$ is a scalar function that describes the magnitude of the noise $\mathbf{z}$, and $\alpha(t) : [0, 1] \rightarrow [0, 1]$ is a scalar function that denotes the magnitude of the data $\mathbf{x}(0)$. The probability density function of $\mathbf{x}(t)$ is denoted as $p_t$.

Two types of SDE are usually considered: the Variance Exploding SDE (VE-SDE) has $\alpha(t) = 1$ for all $t$ and $\sigma(1)$ being a large constant so that $p_1$ is close to $\mathcal{N}(\mathbf{0}, \sigma^2(1)\mathbf{I})$; whereas the Variance Preserving (VP) SDE satisfies $\alpha^2(t) + \sigma^2(t) = 1$ for all $t$ with $\alpha(t) \rightarrow 0$ as $t \rightarrow 1$ so that $p_1$ equals to $\mathcal{N}(\mathbf{0}, \mathbf{I})$. Both VE and VP SDE transform the data distribution to random Gaussian noise as $t$ goes from 0 to 1. For brevity, we discuss the details based on VE-SDE for the remainder of the main text, and discuss the VP-SDE procedure in Appendix C. Though possessing slightly different forms and performing differently depending on the image domain, they share the same mathematical intuition.

**Image synthesis with VE-SDE.** Under these definitions, we can pose the image synthesis problem as gradually removing noise from a noisy observation $\mathbf{x}(t)$ to recover $\mathbf{x}(0)$. This can be performed via a reverse SDE (Anderson, 1982; Song et al., 2021) that travels from $t = 1$ to $t = 0$, based on the knowledge about the noise-perturbed score function $\nabla_{\mathbf{x}} \log p_t(\mathbf{x})$. For example, the sampling procedure for VE-SDE is defined by the following (reverse) SDE:

$$\mathrm{d}\mathbf{x}(t) = \left[ -\frac{\mathrm{d}[\sigma^2(t)]}{\mathrm{d}t} \nabla_{\mathbf{x}} \log p_t(\mathbf{x}) \right] \mathrm{d}t + \sqrt{\frac{\mathrm{d}[\sigma^2(t)]}{\mathrm{d}t}} \mathrm{d}\bar{\mathbf{w}}, \quad (2)$$

where $\bar{\mathbf{w}}$ is a Wiener process when time flows backwards from $t = 1$ to $t = 0$. If we set the initial conditions $\mathbf{x}(1) \sim p_1 = \mathcal{N}(\mathbf{0}, \sigma^2(1)\mathbf{I})$, then the solution to $\mathbf{x}(0)$ will be distributed as $p_{\text{data}}$. In practice, the noise-perturbed score function can be learned through denoising score matching (Vincent, 2011). Denote the learned score model as $\boldsymbol{s}_{\boldsymbol{\theta}}(\mathbf{x}(t), t)$, the learning objective for time $t$ is:

$$L_t = \mathbb{E}_{\mathbf{x}(0) \sim p_{\text{data}}, \mathbf{z} \sim \mathcal{N}(\mathbf{0}, \mathbf{I})}[\|\sigma_t \boldsymbol{s}_{\boldsymbol{\theta}}(\mathbf{x}(t), t) - \mathbf{z}\|_2^2], \quad (3)$$

where $p_{\text{data}}$ is the data distribution and $\mathbf{x}(t)$ is defined as in Equation 1. The overall training objective is a weighted sum over $t$ of each individual learning objective $L_t$, and various weighting procedures have been discussed in Ho et al. (2020); Song et al. (2020; 2021).

With a parametrized score model $\boldsymbol{s}_{\boldsymbol{\theta}}(\mathbf{x}(t), t)$ to approximate $\nabla_{\mathbf{x}} \log p_t(\mathbf{x})$, the SDE solution can be approximated with the Euler-Maruyama method; an update rule from $(t + \Delta t)$ to $t$ is

$$\mathbf{x}(t) = \mathbf{x}(t + \Delta t) + (\sigma^2(t) - \sigma^2(t + \Delta t))\boldsymbol{s}_{\boldsymbol{\theta}}(\mathbf{x}(t), t) + \sqrt{\sigma^2(t) - \sigma^2(t + \Delta t)}\mathbf{z}. \quad (4)$$

where $\mathbf{z} \sim \mathcal{N}(\mathbf{0}, \mathbf{I})$. We can select a particular discretization of the time interval from 1 to 0, initialize $\mathbf{x}(0) \sim \mathcal{N}(\mathbf{0}, \sigma^2(1)\mathbf{I})$ and iterate via Equation 4 to produce an image $\mathbf{x}(0)$.

## 3 GUIDED IMAGE SYNTHESIS AND EDITING WITH SDEDIT

In this section, we introduce SDEdit and describe how we can perform guided image synthesis and editing through an SDE model pretrained on unlabeled images.

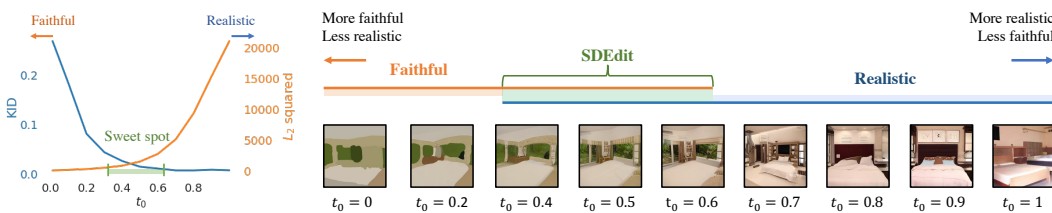

(a) KID and $L_2$ norm **squared** plot with respect to $t_0$.

(b) We illustrate synthesized images of SDEdit with various $t_0$ initializations. $t_0 = 0$ indicates the guide itself, whereas $t_0 = 1$ indicates a random sample.

Figure 3: Trade-off between faithfulness and realism for stroke-based generation on LSUN. As $t_0$ increases, the generated images become **more realistic** while **less faithful**. Given an input, SDEdit aims at generating an image that is both faithful and realistic, which means that we should choose $t_0$ appropriately ($t_0 \in [0.3, 0.6]$ in this example).

**Setup.** The user provides a full resolution image $\mathbf{x}^{(g)}$ in a form of manipulating RGB pixels, which we call a "*guide*". The guide may contain different levels of details; a high-level guide contains only coarse colored strokes, a mid-level guide contains colored strokes on a real image, and a low-level guide contains image patches on a target image. We illustrate these guides in Fig. 1, which can be easily provided by non-experts. Our goal is to produce full resolution images with two desiderata:

**Realism.** The image should appear realistic (*e.g.*, measured by humans or neural networks).

**Faithfulness.** The image should be similar to the guide $\mathbf{x}^{(g)}$ (*e.g.*, measured by $L_2$ distance).

We note that realism and faithfulness are not positively correlated, since there can be realistic images that are not faithful (*e.g.*, a random realistic image) and faithful images that are not realistic (*e.g.*, the guide itself). Unlike regular inverse problems, we do not assume knowledge about the measurement function (*i.e.*, the mapping from real images to user-created guides in RBG pixels is unknown), so techniques for solving inverse problems with score-based models (Dhariwal & Nichol, 2021; Kawar et al., 2021) and methods requiring paired datasets (Isola et al., 2017; Zhu et al., 2017) do not apply here.

**Procedure.** Our method, SDEdit, uses the fact that the reverse SDE can be solved not only from $t_0 = 1$, but also from any intermediate time $t_0 \in (0, 1)$ – an approach not studied by previous SDE-based generative models. We need to find a proper initialization from our guides from which we can solve the reverse SDE to obtain desirable, realistic, and faithful images. For any given guide $\mathbf{x}^{(g)}$, we define the SDEdit procedure as follows:

$$\text{Sample } \mathbf{x}^{(g)}(t_0) \sim \mathcal{N}(\mathbf{x}^{(g)}; \sigma^2(t_0)\mathbf{I}), \text{ then produce } \mathbf{x}(0) \text{ by iterating Equation 4.}$$

We use $\text{SDEdit}(\mathbf{x}^{(g)}; t_0, \theta)$ to denote the above procedure. Essentially, SDEdit selects a particular time $t_0$, add Gaussian noise of standard deviation $\sigma^2(t_0)$ to the guide $\mathbf{x}^{(g)}$ and then solves the corresponding reverse SDE at $t = 0$ to produce the synthesized $\mathbf{x}(0)$.

Apart from the discretization steps taken by the SDE solver, the key hyperparameter for SDEdit is $t_0$, the time from which we begin the image synthesis procedure in the reverse SDE. In the following, we describe a realism-faithfulness trade-off that allows us to select reasonable values of $t_0$.

**Realism-faithfulness trade-off.** We note that for properly trained SDE models, there is a realism-faithfulness trade-off when choosing different values of $t_0$. To illustrate this, we focus on the LSUN dataset, and use high-level stroke paintings as guides to perform stroke-based image generation. We provide experimental details in Appendix D.2. We consider different choices of $t_0 \in [0, 1]$ for the same input. To quantify realism, we adopt neural methods for comparing image distributions, such as the Kernel Inception Score (KID; Bińkowski et al., 2018). If the KID between synthesized images and real images are low, then the synthesized images are realistic. For faithfulness, we measure the squared $L_2$ distance between the synthesized images and the guides $\mathbf{x}^{(g)}$. From Fig. 3, we observe increased realism but decreased faithfulness as $t_0$ increases.

The realism-faithfulness trade-off can be interpreted from another angle. If the guide is far from any realistic images, then we must tolerate at least a certain level of deviation from the guide (non-

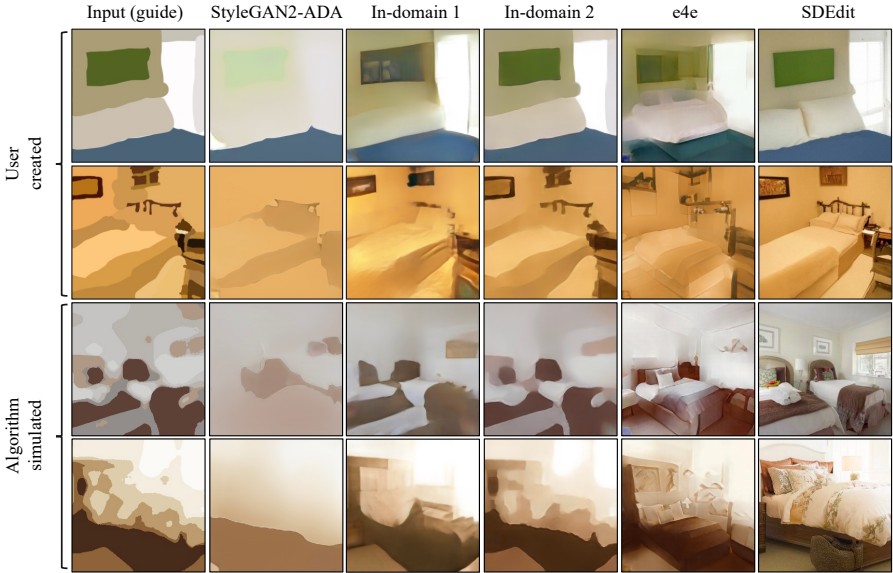

Figure 4: SDEdit generates more realistic and faithful images than state-of-the-art GAN-based models on stroke-based generation (LSUN bedroom). The guide in the first two rows are created by human and the ones in the last two rows are simulated by algorithm.

faithfulness) in order to produce a realistic image. This is illustrated in the following proposition.

**Proposition 1.** *Assume that $\|s_\theta(\mathbf{x}, t)\|_2^2 \leq C$ for all $\mathbf{x} \in \mathcal{X}$ and $t \in [0, 1]$. Then for all $\delta \in (0, 1)$ with probability at least $(1 - \delta)$,*

$$\left\|\mathbf{x}^{(g)} - \mathrm{SDEdit}(\mathbf{x}^{(g)}; t_0, \theta)\right\|_2^2 \leq \sigma^2(t_0)(C\sigma^2(t_0) + d + 2\sqrt{-d \cdot \log \delta} - 2\log \delta) \quad (5)$$

*where $d$ is the number of dimensions of $\mathbf{x}^{(g)}$.*

We provide the proof in Appendix A. On a high-level, the difference from the guides and the synthesized images can be decomposed into the outputs of the score and random Gaussian noise; both would increase as $t_0$ increases, and thus the difference becomes greater. The above proposition suggests that for the image to be realistic with high probability, we must have a large enough $t_0$. On the flip side, if $t_0$ is too large, then the faithfulness to the guide deteriorates, and SDEdit will produce random realistic images (with the extreme case being unconditional image synthesis).

**Choice of $t_0$.** We note that the quality of the guide may affect the overall quality of the synthesized image. For reasonable guides, we find that $t_0 \in [0.3, 0.6]$ works well. However, if the guide is an image with only white pixels, then even the closest "realistic" samples from the model distribution can be quite far, and we must sacrifice faithfulness for better realism by choosing a large $t_0$. In interactive settings (where user draws a sketch-based guide), we can initialize $t_0 \in [0.3, 0.6]$, synthesize a candidate with SDEdit, and ask the user whether the sample should be more faithful or more realistic; from the responses, we can obtain a reasonable $t_0$ via binary search. In large-scale non-interactive settings (where we are given a set of produced guides), we can perform a similar binary search on a randomly selected image to obtain $t_0$ and subsequently fix $t_0$ for all guides in the same task. Although different guides could potentially have different optimal $t_0$, we empirically observe that the shared $t_0$ works well for all reasonable guides in the same task.

**Detailed algorithm and extensions.** We present the general algorithm for VE-SDE in Algorithm 1. Due to space limit, we describe our detailed algorithm for VP-SDE in Appendix C. Essentially, the algorithm is an Euler-Maruyama method for solving $\mathrm{SDEdit}(\mathbf{x}^{(g)}; t_0, \theta)$. For cases where we wish to keep certain parts of the synthesized images to be identical to that of the guides, we can also introduce an additional channel that masks out parts of the image we do not wish to edit. This is a slight modification to the SDEdit procedure mentioned in the main text, and we discuss the details in Appendix C.2.

---

**Algorithm 1** Guided image synthesis and editing with SDEdit (VE-SDE)

---

**Require:** $\mathbf{x}^{(g)}$ (guide), $t_0$ (SDE hyper-parameter), $N$ (total denoising steps)

$\quad \Delta t \leftarrow \frac{t_0}{N}$

$\quad \mathbf{z} \sim \mathcal{N}(\mathbf{0}, \boldsymbol{I})$

$\quad \mathbf{x} \leftarrow \mathbf{x} + \sigma(t_0)\mathbf{z}$

$\quad$ **for** $n \leftarrow N$ **to** $1$ **do**

$\quad\quad t \leftarrow t_0 \frac{n}{N}$

$\quad\quad \mathbf{z} \sim \mathcal{N}(\mathbf{0}, \boldsymbol{I})$

$\quad\quad \epsilon \leftarrow \sqrt{\sigma^2(t) - \sigma^2(t - \Delta t)}$

$\quad\quad \mathbf{x} \leftarrow \mathbf{x} + \epsilon^2 \boldsymbol{s_\theta}(\mathbf{x}, t) + \epsilon \mathbf{z}$

$\quad$ **end for**

$\quad$ **Return** $\mathbf{x}$

---

# 4 RELATED WORK

**Conditional GANs.** Conditional GANs for image editing (Isola et al., 2017; Zhu et al., 2017; Jo & Park, 2019; Liu et al., 2021) learn to directly generate an image based on a user input, and have demonstrated success on a variety of tasks including image synthesis and editing (Portenier et al., 2018; Chen & Koltun, 2017; Dekel et al., 2018; Wang et al., 2018; Park et al., 2019; Zhu et al., 2020b; Jo & Park, 2019; Liu et al., 2021), inpainting (Pathak et al., 2016; Iizuka et al., 2017; Yang et al., 2017; Liu et al., 2018), photo colorization (Zhang et al., 2016; Larsson et al., 2016; Zhang et al., 2017; He et al., 2018), semantic image texture and geometry synthesis (Zhou et al., 2018; Guérin et al., 2017; Xian et al., 2018). They have also achieved strong performance on image editing using user sketch or color (Jo & Park, 2019; Liu et al., 2021; Sangkloy et al., 2017). However, conditional models have to be trained on both original and edited images, thus requiring data collection and model re-training for new editing tasks. Thus, applying such methods to on-the-fly image manipulation is still challenging since a new model needs to be trained for each new application. Unlike conditional GANs, SDEdit only requires training on the original image. As such, it can be directly applied to various editing tasks at test time as illustrated in Fig. 1.

**GANs inversion and editing.** Another mainstream approach to image editing involves GAN inversion (Zhu et al., 2016; Brock et al., 2017), where the input is first projected into the latent space of an unconditional GAN before synthesizing a new image from the modified latent code. Several methods have been proposed in this direction, including fine-tuning network weights for each image (Bau et al., 2019a; Pan et al., 2020; Roich et al., 2021), choosing better or multiple layers to project and edit (Abdal et al., 2019; 2020; Gu et al., 2020; Wu et al., 2021), designing better encoders (Richardson et al., 2021; Tov et al., 2021), modeling image corruption and transformations (Anirudh et al., 2020; Huh et al., 2020), and discovering meaningful latent directions (Shen et al., 2020; Goetschalckx et al., 2019; Jahanian et al., 2020; Härkönen et al., 2020). However, these methods need to define different loss functions for different tasks. They also require GAN inversion, which can be inefficient and inaccurate for various datasets (Huh et al., 2020; Karras et al., 2020b; Bau et al., 2019b; Xu et al., 2021).

**Other generative models.** Recent advances in training non-normalized probabilistic models, such as score-based generative models (Song & Ermon, 2019; 2020; Song et al., 2021; Ho et al., 2020; Song et al., 2020; Jolicoeur-Martineau et al., 2021) and energy-based models (Ackley et al., 1985; Gao et al., 2017; Du & Mordatch, 2019; Xie et al., 2018; 2016; Song & Kingma, 2021), have achieved comparable image sample quality as GANs. However, most of the prior works in this direction have focused on unconditional image generation and density estimation, and state-of-the-art techniques for image editing and synthesis are still dominated by GAN-based methods. In this work, we focus on the recently emerged generative modeling with stochastic differential equations (SDE), and study its application to controllable image editing and synthesis tasks. A concurrent work (Choi et al., 2021) performs conditional image synthesis with diffusion models, where the conditions can be represented as the known function of the underlying true image.

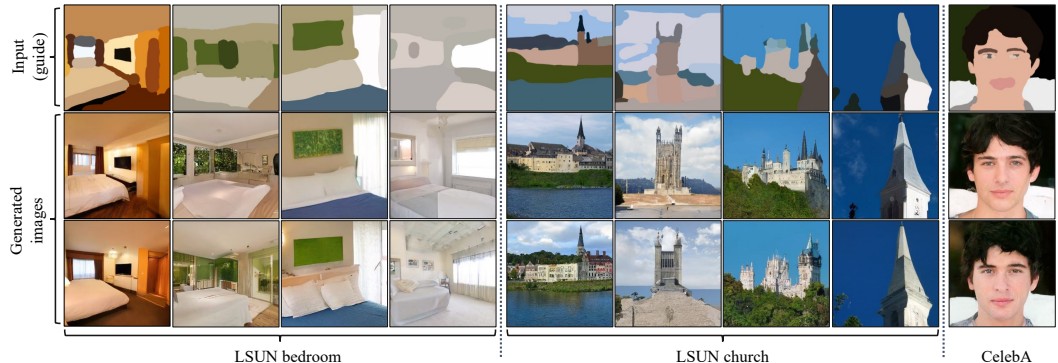

Figure 5: SDEdit can generate realistic, faithful and diverse images for a given stroke input drawn by human.

## 5 EXPERIMENTS

In this section, we show that SDEdit is able to outperform state-of-the-art GAN-based models on stroke-based image synthesis and editing as well as image compositing. Both SDEdit and the baselines use publicly available pre-trained checkpoints. Based on the availability of open-sourced SDE checkpoints, we use VP-SDE for experiments on LSUN datasets, and VE-SDE for experiments on CelebA-HQ.

| Baselines | Faithfulness score ($L_2$) ↓ | SDEdit is more realistic (MTurk) ↑ | SDEdit is more satisfactory (Mturk) ↑ |
|---|---|---|---|
| In-domain GAN-1 | 101.18 | 94.96% | 89.48% |
| In-domain GAN-2 | 57.11 | 97.87% | 89.51% |
| StyleGAN2-ADA | 68.12 | 98.09% | 91.72% |
| e4e | 53.76 | 80.34% | 75.43% |
| SDEdit | **32.55** | – | – |

Table 1: SDEdit outperforms all the GAN baselines on stroke-based generation on LSUN (bedroom). The input strokes are created by human users. The rightmost two columns stand for the percentage of MTurk workers that prefer SDEdit to the baseline for pairwise comparison.

**Evaluation metrics.** We evaluate the editing results based on *realism* and *faithfulness*. To quantify *realism*, we use Kernel Inception Score (KID) between the generated images and the target realistic image dataset (details in Appendix D.2), and pairwise human evaluation between different approaches with Amazon Mechanical Turk (MTurk). To quantify *faithfulness*, we report the $L_2$ distance summed over all pixels between the guide and the edited output image normalized to [0,1]. We also consider LPIPS (Zhang et al., 2018) and MTurk human evaluation for certain experiments. To quantify the overall human satisfaction score (*realism + faithfulness*), we leverage MTurk human evaluation to perform pairwise comparsion between the baselines and SDEdit (see Appendix F).

### 5.1 STROKE-BASED IMAGE SYNTHESIS

Given an input stroke painting, our goal is to generate a *realistic* and *faithful* image *when no paired data is available*. We consider stroke painting guides created by human users (see Fig. 5). At the same time, we also propose an algorithm to automatically simulate user stroke paintings based on a source image (see Fig. 4), allowing us to perform large scale quantitative evaluations for SDEdit. We provide more details in Appendix D.2.

**Baselines.** For comparison, we choose three state-of-the-art GAN-based image editing and synthesis methods as our baselines. Our first baseline is the image projection method used in StyleGAN2-ADA[1] (Karras et al., 2020a), where inversion is done in the $W^+$ space of StyleGANs by minimizing the perceptual loss. Our second baseline is in-domain GAN[2] (Zhu et al., 2020a), where inversion is accomplished by running optimization steps on top of an encoder. Specifically, we consider two versions of the in-domain GAN inversion techniques: the first one (denoted as In-domain GAN-1)

---

[1] https://github.com/NVlabs/stylegan2-ada
[2] https://github.com/genforce/idinvert_pytorch

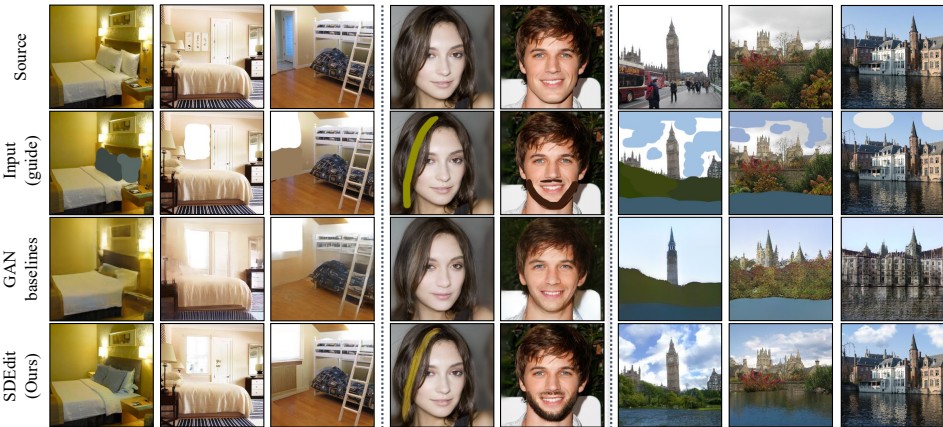

Figure 6: Stroke-based image editing with SDEdit on LSUN bedroom, CelebA-HQ, and LSUN church datasets. For comparison, we show the results of GAN baselines, where results for LSUN bedroom and CelebA-HQ are obtained by in-domain GAN (the leftmost 5 panels), and results for LSUN church are from StyleGAN2-ADA (the rightmost 3 panels). We observe that SDEdit is able to produce more faithful and realistic editing compared to the baselines.

only uses the encoder to maximize the inversion speed, whereas the second (denoted as In-domain GAN-2) runs additional optimization steps to maximize the inversion accuracy. Our third baseline is e4e[3] (Tov et al., 2021), whose encoder objective is explicitly designed to balance between perceptual quality and editability by encouraging to invert images close to $W$ space of a pretrained StyleGAN model.

**Results.** We present qualitative comparison results in Fig. 4. We observe that all baselines struggle to generate realistic images based on stroke painting inputs whereas SDEdit successfully generates realistic images that preserve semantics of the input stroke painting. As shown in Fig. 5, SDEdit can also synthesize diverse images for the same input. We present quantitative comparison results using user-created stroke guides in Table 1 and algorithm-simulated stroke guides in Table 2. We report the $L_2$ distance for faithfulness comparison, and leverage MTurk (see Appendix F) or KID scores for realism comparison. To quantify the overall human satisfaction score (faithfulness + realism), we ask a different set of MTurk workers to perform another 3000 pair-

| Methods | LSUN Bedroom | | LSUN Church | |
|---|---|---|---|---|
| | $L_2 \downarrow$ | KID $\downarrow$ | $L_2 \downarrow$ | KID $\downarrow$ |
| In-domain GAN-1 | 105.23 | 0.1147 | - | - |
| In-domain GAN-2 | 76.11 | 0.2070 | - | - |
| StyleGAN2-ADA | 74.03 | 0.1750 | 72.41 | 0.1544 |
| e4e | 52.40 | 0.0464 | 68.53 | 0.0354 |
| SDEdit (ours) | **36.76** | **0.0030** | **37.67** | **0.0156** |

Table 2: SDEdit outperforms all the GAN baselines on both faithfulness and realism for stroke-based image generation. The input strokes are generated with the stroke-simulation algorithm. KID is computed using the generated images and the corresponding validation sets (see Appendix D.2).

wise comparisons between the baselines and SDEdit based on *both faithfulness* and *realism*. We observe that SDEdit outperforms GAN baselines on all the evaluation metrics, beating the baselines by more than **80%** on realism scores and **75%** on overall satisfaction scores. We provide more experimental details in Appendix C and more results in Appendix E.

## 5.2 FLEXIBLE IMAGE EDITING

In this section, we show that SDEdit is able to outperform existing GAN-based models on image editing tasks. We focus on LSUN (bedroom, church) and CelebA-HQ datasets, and provide more details on the experimental setup in the Appendix D.

**Stroke-based image editing.** Given an image with stroke edits, we want to generate a realistic and faithful image based on the user edit. We consider the same GAN-based baselines (Zhu et al., 2020a; Karras et al., 2020a; Tov et al., 2021) as our previous experiment. As shown in Fig. 6,

---

[3] https://github.com/omertov/encoder4editing

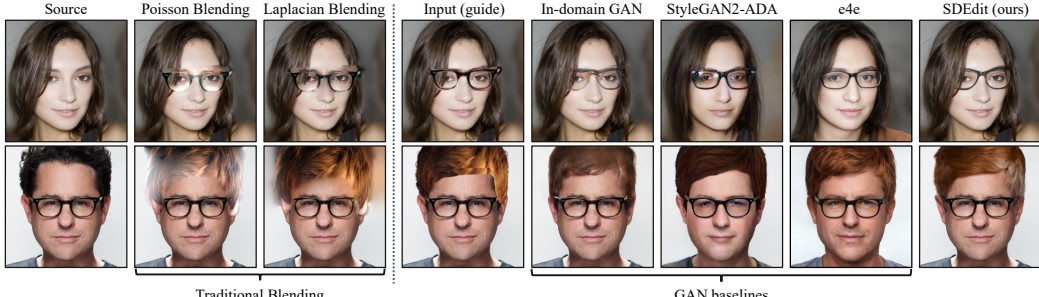

Figure 7: SDEdit is able to achieve realistic while more faithful editing results compared to traditional blending and recent GAN-based approaches for image compositing on CelebA-HQ. Quantitative results are reported in Table 3.

| Methods | $L_2 \downarrow$ (faithfulness) | SDEdit more realistic (Mturk) $\uparrow$ | SDEdit more satisfactory (Mturk) $\uparrow$ | LPIPS (masked) $\downarrow$ |
|---|---|---|---|---|
| Laplacian Blending | 68.45 | 75.27% | 83.73% | 0.09 |
| Poisson Blending | 63.04 | 75.60% | 82.18% | 0.05 |
| In-domain GAN | 36.67 | 53.08% | 73.53% | 0.23 |
| StyleGAN2-ADA | 69.38 | 74.12% | 83.43% | 0.21 |
| e4e | 53.90 | 43.67% | 66.00% | 0.33 |
| SDEdit (ours) | **21.70** | – | – | **0.03** |

Table 3: Image compositing experiments on CelebA-HQ. The middle two columns indicate the percentage of MTurk workers that prefer SDEdit. We also report the masked LPIPS distance between edited and unchanged images to quantify undesired changes outside the masks. We observe that SDEdit is able to achieve realistic editing while being more faithful than the baselines, beating the baseline by up to **83.73%** on overall satisfaction score by human evaluators.

results generated by the baselines tend to introduce undesired modifications, occasionally making the region outside the stroke blurry. In contrast, SDEdit generates image edits that are *both realistic* and *faithful* to the input, while avoiding making undesired modifications. We provide extra results in Appendix E.

**Image compositing.** We focus on compositing images on the CelebA-HQ dataset (Karras et al., 2017). Given an image randomly sampled from the dataset, we ask users to specify how they want the edited image to look like using pixel patches copied from other reference images as well as the pixels they want to perform modifications (see Fig. 7). We compare our method with traditional blending algorithms (Burt & Adelson, 1987; Pérez et al., 2003) and the same GAN baselines considered previously. We perform qualitative comparison in Fig. 7. For quantitative comparison, we report the $L_2$ distance to quantify faithfulness. To quantify realism, we ask MTurk workers to perform 1500 pairwise comparisons between the baselines and SDEdit. To quantify user satisfaction score (faithfulness + realism), we ask different workers to perform another 1500 pairwise comparisons against SDEdit. To quantify undesired changes (*e.g.* change of identity), we follow Bau et al. (2020) to compute masked LPIPS (Zhang et al., 2018). As evidenced in Table 3, we observe that SDEdit is able to generate both faithful and realistic images with much better LPIPS scores than the baselines, outperforming the baselines by up to **83.73%** on overall satisfaction score and **75.60%** on realism. Although our realism score is marginally lower than e4e, images generated by SDEdit are more faithful and more satisfying overall. We present more experiment details in Appendix D.

## 6 CONCLUSION

We propose Stochastic Differential Editing (SDEdit), a guided image editing and synthesis method via generative modeling of images with stochastic differential equations (SDEs) allowing for balanced realism and faithfulness. Unlike image editing techniques via GAN inversion, our method does not require task-specific optimization algorithms for reconstructing inputs, and is particularly suitable for datasets or tasks where GAN inversion losses are hard to design or optimize. Unlike conditional GANs, our method does not require collecting new datasets for the "guide" images or re-training models, both of which could be expensive or time-consuming. We demonstrate that SDEdit outperforms existing GAN-based methods on a variety of image synthesis and editing tasks.

**Acknowledgments.** The authors would like to thank Kristy Choi for proofreading. This research was in part supported by NSF (#1651565, #1522054, #1733686), ONR (N00014-19-1-2145), AFOSR (FA9550-19-1-0024), ARO (W911NF-15-1-0479), Autodesk, Google, Bosch, Stanford Institute for Human-Centered AI (HAI), Stanford Center for Integrated Facility Engineering (CIFE), Amazon Research Award (ARA), and Amazon AWS. Yang Song is supported by the Apple PhD Fellowship in AI/ML. J.-Y. Zhu is partly supported by Naver Corporation.

## ETHICS STATEMENT

In this work, we propose SDEdit, which is a new image synthesis and editing methods based on generative stochastic differential equations (SDEs). In our experiments, all the considered datasets are open-sourced and publicly available, being used under permission. Similar to commonly seen deep-learning based image synthesis and editing algorithms, our method has both positive and negative societal impacts depending on the applications and usages. On the positive side, SDEdit enables everyday users with or without artistic expertise to create and edit photo-realistic images with minimum effort, lowering the barrier to entry for visual content creation. On the other hand, SDEdit can be used to generate high-quality edited images that are hard to be distinguished from real ones by humans, which could be used in malicious ways to deceive humans and spread misinformation. Similar to commonly seen deep-learning models (such as GAN-based methods for face-editing), SDEdit might be exploited by malicious users with potential negative impacts. In our code release, we will explicitly specify allowable uses of our system with appropriate licenses.

We also notice that forensic methods for detecting fake machine-generated images mostly focus on distinguishing samples generated by GAN-based approaches. Due to the different underlying nature between GANs and generative SDEs, we observe that state-of-the-art approaches for detecting fake images generated by GANs (Wang et al., 2020) struggle to distinguish fake samples generated by SDE-based models. For instance, on the LSUN bedroom dataset, it only successfully detects less than 3% of SDEdit-generated images whereas being able to distinguish up to 93% on GAN-based generation. Based on these observations, we believe developing forensic methods for SDE-based models is also critical as SDE-based methods become more prevalent.

For human evaluation experiments, we leveraged Amazon Mechanical Turk (MTurk). For each worker, the evaluation HIT contains 15 pairwise comparison questions for comparing edited images. The reward per task is kept as 0.2$. Since each task takes around 1 minute, the wage is around 12$ per hour. We provide more details on Human evaluation experiments in Appendix F. We also note that the bias of human evaluators (MTurk workers) and the bias of users (through the input "guidance") could potentially affect the evaluation metrics and results used to track the progress towards guided image synthesis and editing.

## REPRODUCIBILITY STATEMENT

1. Our code is released at `https://github.com/ermongroup/SDEdit`.
2. We use open source datasets and SDE checkpoints on the corresponding datasets. We did not train any SDE models.
3. Proofs are provided in Appendix A.
4. Extra details on SDEdit and pseudocode are provided in Appendix C.
5. Details on experimental settings are provided in Appendix D.
6. Extra experimental results are provided in Appendix E.
7. Details on human evaluation are provided in Appendix F.

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

## A    Proofs

**Proposition 1.** *Assume that $\|s_\theta(\mathbf{x}, t)\|_2^2 \leq C$ for all $\mathbf{x} \in \mathcal{X}$ and $t \in [0, 1]$. Then for all $\delta \in (0, 1)$ with probability at least $(1 - \delta)$,*

$$\left\| \mathbf{x}^{(g)} - \text{SDEdit}(\mathbf{x}^{(g)}; t_0, \theta) \right\|_2^2 \leq \sigma^2(t_0)(C\sigma^2(t_0) + d + 2\sqrt{-d \cdot \log \delta} - 2\log \delta) \qquad (5)$$

*where $d$ is the number of dimensions of $\mathbf{x}^{(g)}$.*

*Proof.* Denote $\mathbf{x}^{(g)}(0) = \text{SDEdit}(\mathbf{x}^{(g)}; t, \theta)$, then

$$\left\| \mathbf{x}^{(g)}(t_0) - \mathbf{x}^{(g)}(0) \right\|_2^2 = \left\| \int_{t_0}^0 \frac{d\mathbf{x}^{(g)}(t)}{dt} dt \right\|_2^2 \qquad (6)$$

$$= \left\| \int_{t_0}^0 \left[ -\frac{d[\sigma^2(t)]}{dt} s_\theta(\mathbf{x}, t; \theta) \right] dt + \sqrt{\frac{d[\sigma^2(t)]}{dt}} d\bar{\mathbf{w}} \right\|_2^2 \qquad (7)$$

$$\leq \left\| \int_{t_0}^0 \left[ -\frac{d[\sigma^2(t)]}{dt} s_\theta(\mathbf{x}, t; \theta) \right] dt \right\|_2^2 + \left\| \int_{t_0}^0 \sqrt{\frac{d[\sigma^2(t)]}{dt}} d\bar{\mathbf{w}} \right\|_2^2 \qquad (8)$$

From the assumption over $s_\theta(\mathbf{x}, t; \theta)$, the first term is not greater than

$$C \left\| \int_{t_0}^0 \left[ -\frac{d[\sigma^2(t)]}{dt} \right] dt \right\|_2^2 = C\sigma^4(t_0),$$

where equality could only happen when each score output has a squared $L_2$ norm of $C$ and they are linearly dependent to one other. The second term is independent to the first term as it only concerns random noise; this is equal to the squared $L_2$ norm of a random variable from a Wiener process at time $t = 0$, with marginal distribution being $\epsilon \sim \mathcal{N}(\mathbf{0}, \sigma^2(t_0)\mathbf{I})$ (this marginal does not depend on the discretization steps in Euler-Maruyama). The squared $L_2$ norm of $\epsilon$ divided by $\sigma^2(t_0)$ is a $\chi^2$-distribution with $d$-degrees of freedom. From Laurent & Massart (2000), Lemma 1, we have the following one-sided tail bound:

$$\Pr(\|\epsilon\|_2^2 / \sigma^2(t_0) \geq d + 2\sqrt{d \cdot -\log \delta} - 2\log \delta) \leq \exp(\log \delta) = \delta. \qquad (9)$$

Therefore, with probability at least $(1 - \delta)$, we have that:

$$\left\| \mathbf{x}^{(g)}(t_0) - \mathbf{x}^{(g)}(0) \right\|_2^2 \leq \sigma^2(t_0)(C\sigma^2(t_0) + d + 2\sqrt{-d \cdot \log \delta} - 2\log \delta), \qquad (10)$$

completing the proof. $\square$

## B    Extra ablation studies

In this section, we perform extra ablation studies and analysis for SDEdit.

### B.1    Analysis on the quality of user guide

As discussed in Section 3, if the guide is far from any realistic images (*e.g.*, random noise or has an unreasonable composition) , then we must tolerate at least a certain level of deviation from the guide (non-faithfulness) in order to produce a realistic image.

For practical applications, we perform extra ablation studies on how the quality of guided stroke would affect the results in Fig. 8, Fig. 9 and Table 4. Specifically, in Fig. 8 we consider stroke input of 1) a human face with limited detail for a CelebA-HQ model, 2) a human face with spikes for a CelebA-HQ model, 3) a building with limited detail for a LSUN-church model, 4) a horse for a LSUN-church model. We observe that SDEdit is in general tolerant to different kinds of user inputs. In Table 4, we quantitatively analyze the effect of user guide quality using simulated stroke paintings as input. Described in Appendix D.2, the human-stroke-simulation algorithm uses

different numbers of colors to generate stroke guides with different levels of detail. We compare SDEdit with baselines qualitatively in Fig. 9 and quantitatively in Table 4. Similarly, we observe that SDEdit has a high tolerance to input guides and consistently outperforms the baselines across all setups in this experiment.

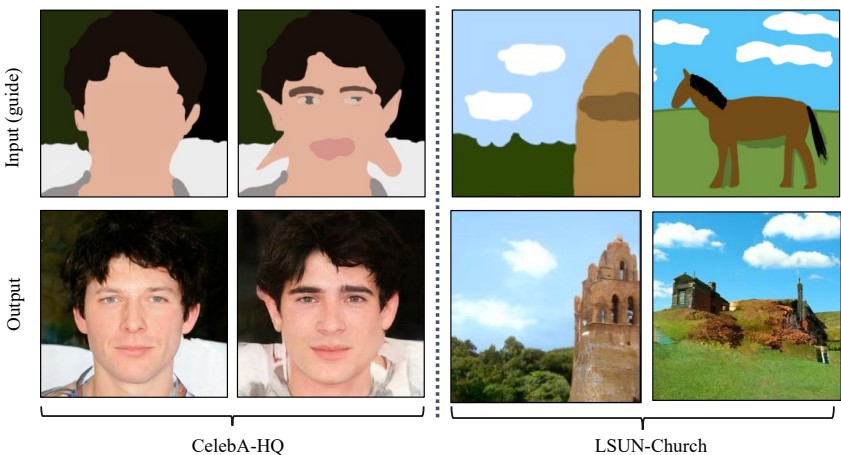

Figure 8: Analysis on the quality of user guide for stoke-based image synthesis. We observe that SDEdit is in general tolerant to different kinds of user inputs.

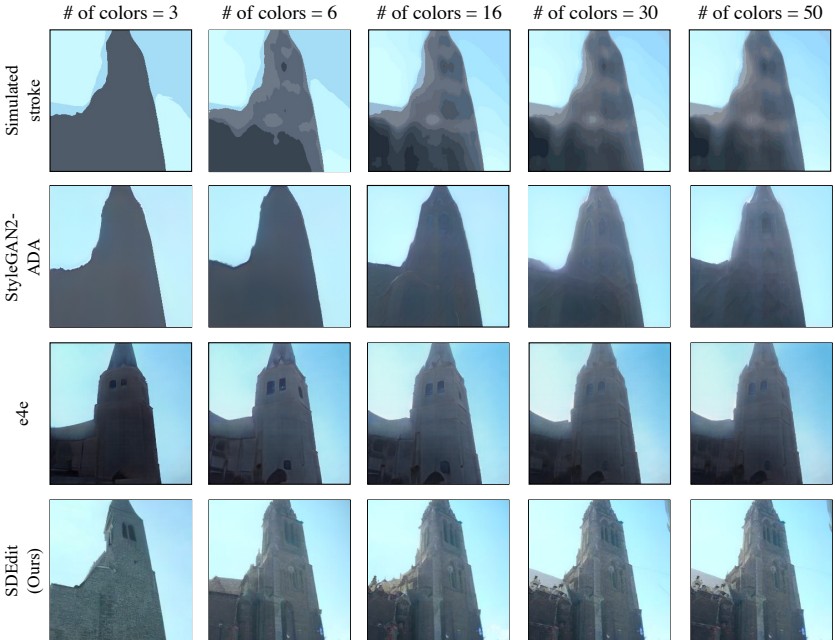

Figure 9: Analysis on the quality of user guide for stoke-based image synthesis. We observe that SDEdit is in general tolerant to different kinds of user inputs.

| # of colors | StyleGAN2-ADA | | e4e | | SDEdit (Ours) | |
|---|---|---|---|---|---|---|
| | KID $\downarrow$ | $L_2 \downarrow$ | KID $\downarrow$ | $L_2 \downarrow$ | KID $\downarrow$ | $L_2 \downarrow$ |
| 3 | 0.1588 | 67.22 | 0.0379 | 70.73 | **0.0233** | **36.00** |
| 6 | 0.1544 | 72.41 | 0.0354 | 68.53 | **0.0156** | **37.67** |
| 16 | 0.0923 | 69.52 | 0.0319 | 68.20 | **0.0135** | **37.70** |
| 30 | 0.0911 | 67.11 | 0.0304 | 68.66 | **0.0128** | **37.42** |
| 50 | 0.0922 | 65.28 | 0.0307 | 68.80 | **0.0126** | **37.40** |

Table 4: We compare SDEdit with baselines quantitatively on LSUN-church dataset on stroke-based generation. "# of colors" denotes the number of colors used to generate the synthetic stroke paintings, with fewer colors corresponding to a less accurate and less detailed input guide (see Fig. 9). We observe that SDEdit consistently achieves more realistic and more faithful outputs and outperforms the baselines across all setups.

## B.2 FLEXIBLE IMAGE EDITING WITH SDEDIT

In this section, we perform extra image editing experiments including editing closing eyes Fig. 10, opening mouth, and changing lip color Fig. 11. We observe that SDEdit can still achieve reasonable editing results, which shows that SDEdit is capable of flexible image editing tasks.

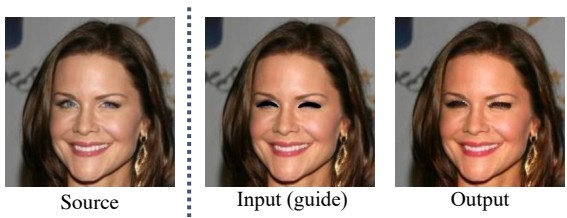

Source      Input (guide)      Output

Figure 10: Flexible image editing on closing eyes with SDEdit.

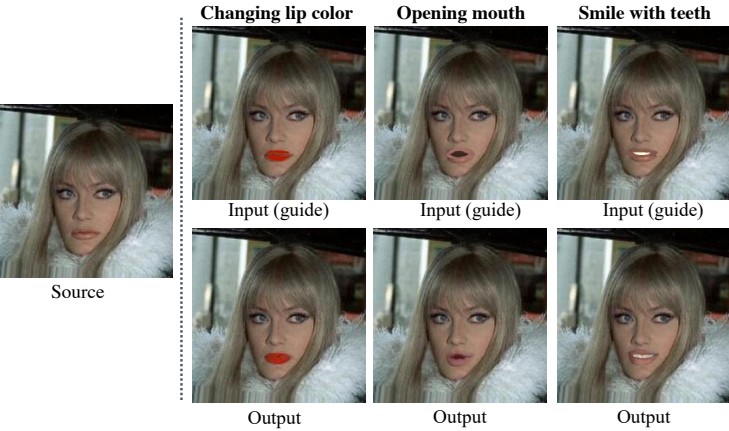

Figure 11: Flexible image editing on mouth with SDEdit.

## B.3 ANALYSIS ON $t_0$

In this section, we provide extra analysis on the effect of $t_0$ (see Fig. 12). As illustrated in Fig. 3, we can tune $t_0$ to tradeoff between faithfulness and realism—with a smaller $t_0$ corresponding to a more faithful but less realistic generated image. If we want to keep the brown stroke in Fig. 12, we can reduce $t_0$ to increase its faithfulness which could potentially decrease its realism. Additional analysis can be found in Appendix D.2.

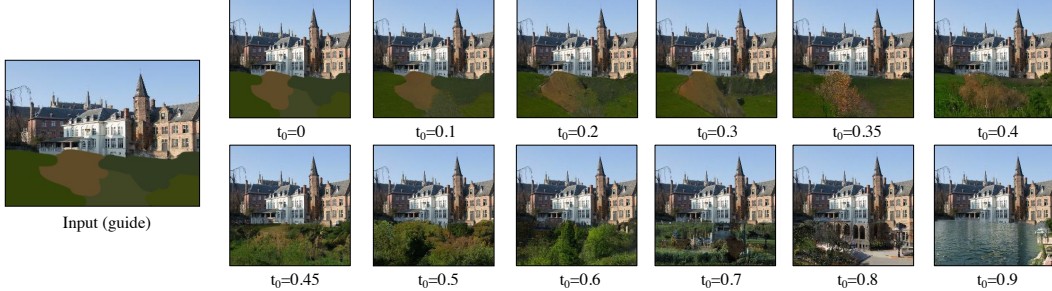

Figure 12: Extra analysis on $t_0$. As $t_0$ increases, the generated images become more realistic while less faithful.

## B.4 EXTRA COMPARISON WITH OTHER BASELINES

We perform extra comparison with SC-FEGAN (Jo & Park, 2019) in Fig. 13. We observe that SDEdit is able to have more realistic results than SC-FEGAN (Jo & Park, 2019) when using the same stroke input guide. We also present results for SC-FEGAN (Jo & Park, 2019) where we use extra sketch together with stroke as the input guide (see Fig. 14). We observe that SDEdit is still able to outperform SC-FEGAN in terms of realism even when SC-FEGAN is using both sketch and stroke as the input guide.

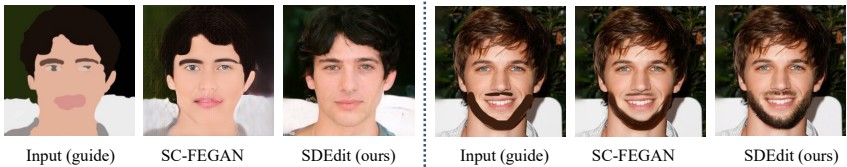

Figure 13: Comparison with SC-FEGAN (Jo & Park, 2019) on stroke-based image synthesis and editing. We observe that SDEdit is able to generate more realistic results than SC-FEGAN.

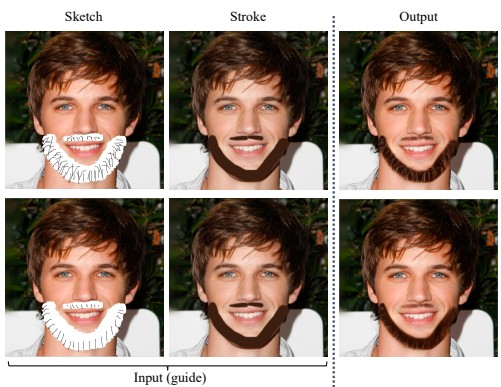

Figure 14: Stroke-based editing for SC-FEGAN (Jo & Park, 2019) using both stroke and extra sketch as the input guide. We observe that SDEdit still outperforms SC-FEGAN using only stroke as the input guide.

## B.5 COMPARISON WITH SONG ET AL. (2021)

Methods proposed by Song et al. (2021) introduce an extra noise-conditioned classifier for conditional generation and the performance of the classifier is critical to the conditional generation performance. Their settings are more similar to regular inverse problems where the measurement function is known, which is discussed in Section 3. Since we do not have a known "measurement" function

for user-generated guides, their approach cannot be directly applied to user-guided image synthesis or editing in the form of manipulating pixel RGB values. To deal with this limitation, SDEdit initializes the reverse SDE based on user input and modifies $t_0$ accordingly—an approach different from Song et al. (2021) (which always have the same initialization). This technique allows SDEdit to achieve faithful and realistic image editing or generation results without extra task-specific model learning (*e.g.*, an additional classifier in Song et al. (2021)).

For practical applications, we compare with Song et al. (2021) on stroke-based image synthesis and editing where we do not learn an extra noise-conditioned classifier (see Fig. 15). In fact, we are also unable to learn the noise-conditioned classifier since we do not have a known "measurement" function for user-generated guides and we only have one random user input guide instead of a dataset of input guide. We observe that this application of Song et al. (2021) fails to generate faithful results by performing random inpainting (see Fig. 15). SDEdit, on the other hand, generates both realistic and faithful images without learning extra task-specific models (*e.g.*, an additional classifier) and can be directly applied to pretrained SDE-based generative models, allowing for guided image synthesis and editing using SDE-based models. We believe this shows the novelty and contribution of SDEdit.

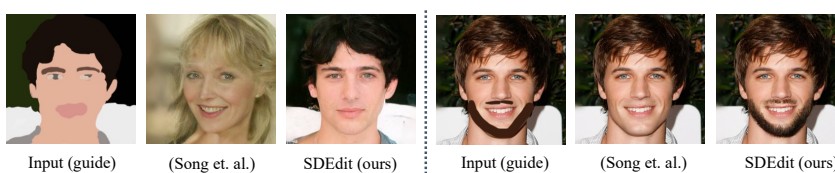

| Input (guide) | (Song et. al.) | SDEdit (ours) | Input (guide) | (Song et. al.) | SDEdit (ours) |

Figure 15: Comparison with Song et al. (2021) on stroke-based image synthesis and editing. We observe that SDEdit is able to generate more faithful results than Song et al. (2021) without training an extra task-specific model (*e.g.*, an additional classifier).

## C  DETAILS ON SDEDIT

### C.1  DETAILS ON THE VP AND VE SDES

We follow the definitions of VE and VP SDEs in Song et al. (2021), and adopt the same settings therein.

**VE-SDE**  In particular, for the VE SDE, we choose

$$\sigma(t) = \begin{cases} 0, & t = 0 \\ \sigma_{\min}\left(\frac{\sigma_{\max}}{\sigma_{\min}}\right)^t, & t > 0 \end{cases}$$

where $\sigma_{\min} = 0.01$ and $\sigma_{\max} = 380, 378, 348, 1348$ for LSUN churches, bedroom, FFHQ/CelebA-HQ $256 \times 256$, and FFHQ $1024 \times 1024$ datasets respectively.

**VP-SDE**  For the VP SDE, it takes the form of

$$d\mathbf{x}(t) = -\frac{1}{2}\beta(t)\mathbf{x}(t)dt + \sqrt{\beta(t)}d\mathbf{w}(t), \tag{11}$$

where $\beta(t)$ is a positive function. In experiments, we follow Song et al. (2021); Ho et al. (2020); Dhariwal & Nichol (2021) and set

$$\beta(t) = \beta_{\min} + t(\beta_{\max} - \beta_{\min}),$$

For SDE trained by Song et al. (2021); Ho et al. (2020) we use $\beta_{\min} = 0.1$ and $\beta_{\max} = 20$; for SDE trained by Dhariwal & Nichol (2021), the model learns to rescale the variance based on the same choices of $\beta_{\min}$ and $\beta_{\max}$. We always have $p_1(\mathbf{x}) \approx \mathcal{N}(\mathbf{0}, \boldsymbol{I})$ under these settings.

Solving the reverse VP SDE is similar to solving the reverse VE SDE. Specifically, we follow the iteration rule below:

$$\mathbf{x}_{n-1} = \frac{1}{\sqrt{1 - \beta(t_n)\Delta t}}(\mathbf{x}_n + \beta(t_n)\Delta t \boldsymbol{s_\theta}(\mathbf{x}(t_n), t_n)) + \sqrt{\beta(t_n)\Delta t}\,\mathbf{z}_n, \tag{12}$$

where $\mathbf{x}_N \sim \mathcal{N}(\mathbf{0}, \boldsymbol{I})$, $\mathbf{z}_n \sim \mathcal{N}(\mathbf{0}, \boldsymbol{I})$ and $n = N, N-1, \cdots, 1$.

C.2    DETAILS ON STOCHASTIC DIFFERENTIAL EDITING

In generation the process detailed in Algorithm 1 can also be repeated for $K$ number of times as detailed in Algorithm 2. Note that Algorithm 1 is a special case of Algorithm 2: when $K = 1$, we recover Algorithm 1. For VE-SDE, Algorithm 2 converts a stroke painting to a photo-realistic image, which typically modifies all pixels of the input. However, in cases such as image compositing and stroke-based editing, certain regions of the input are already photo-realistic and therefore we hope to leave these regions intact. To represent a specific region, we use a binary mask $\mathbf{\Omega} \in \{0, 1\}^{C \times H \times W}$ that evaluates to 1 for editable pixels and 0 otherwise. We can generalize Algorithm 2 to restrict editing in the region defined by $\mathbf{\Omega}$.

For editable regions, we perturb the input image with the forward SDE and generate edits by reversing the SDE, using the same procedure in Algorithm 2. For uneditable regions, we perturb it as usual but design the reverse procedure carefully so that it is guaranteed to recover the input. Specifically, suppose $\mathbf{x} \in \mathbb{R}^{C \times H \times W}$ is an input image of height $H$, width $W$, and with $C$ channels. Our algorithm first perturbs $\mathbf{x}(0) = \mathbf{x}$ with an SDE running from $t = 0$ till $t = t_0$ to obtain $\mathbf{x}(t_0)$. Afterwards, we denoise $\mathbf{x}(t_0)$ with separate methods for $\mathbf{\Omega} \odot \mathbf{x}(t)$ and $(\mathbf{1} - \mathbf{\Omega}) \odot \mathbf{x}(t)$, where $\odot$ denotes the element-wise product and $0 \leq t \leq t_0$. For $\mathbf{\Omega} \odot \mathbf{x}(t)$, we simulate the reverse SDE (Song et al., 2021) and project the results by element-wise multiplication with $\mathbf{\Omega}$. For $(\mathbf{1} - \mathbf{\Omega}) \odot \mathbf{x}(t)$, we set it to $(\mathbf{1} - \mathbf{\Omega}) \odot (\mathbf{x} + \sigma(t)\mathbf{z})$, where $\mathbf{z} \sim \mathcal{N}(\mathbf{0}, \mathbf{I})$. Here we gradually reduce the noise magnitude according to $\sigma(t)$ to make sure $\mathbf{\Omega} \odot \mathbf{x}(t)$ and $(\mathbf{1} - \mathbf{\Omega}) \odot \mathbf{x}(t)$ have comparable amount of noise. Moreover, since $\sigma(t) \to 0$ as $t \to 0$, this ensures that $(\mathbf{1} - \mathbf{\Omega}) \odot \mathbf{x}(t)$ converges to $(\mathbf{1} - \mathbf{\Omega}) \odot \mathbf{x}$, keeping the uneditable part of $\mathbf{x}$ intact. The complete SDEdit method (for VE-SDEs) is given in Algorithm 3. We provide algorithm for VP-SDEs in Algorithm 4 and the corresponding masked version in Algorithm 5.

With different inputs to Algorithm 3 or Algorithm 5, we can perform multiple image synthesis and editing tasks with a single unified approach, including but not limited to the following:

- **Stroke-based image synthesis:** We can recover Algorithm 2 or Algorithm 4 by setting all entries in $\mathbf{\Omega}$ to 1.

- **Stroke-based image editing:** Suppose $\mathbf{x}^{(g)}$ is an image marked by strokes, and $\mathbf{\Omega}$ masks the part that are not stroke pixels. We can reconcile the two parts of $\mathbf{x}^{(g)}$ with Algorithm 3 to obtain a photo-realistic image.

- **Image compositing:** Suppose $\mathbf{x}^{(g)}$ is an image superimposed by elements from two images, and $\mathbf{\Omega}$ masks the region that the users do not want to perform editing, we can perform image compositing with Algorithm 3 or Algorithm 5.

---

**Algorithm 2** Guided image synthesis and editing (VE-SDE)

---

**Require:** $\mathbf{x}^{(g)}$ (guide), $t_0$ (SDE hyper-parameter), $N$ (total denoising steps), $K$ (total repeats)

$\quad \Delta t \leftarrow \frac{t_0}{N}$
$\quad$ **for** $k \leftarrow 1$ **to** $K$ **do**
$\quad\quad \mathbf{z} \sim \mathcal{N}(\mathbf{0}, \mathbf{I})$
$\quad\quad \mathbf{x} \leftarrow \mathbf{x} + \sigma(t_0)\mathbf{z}$
$\quad\quad$ **for** $n \leftarrow N$ **to** $1$ **do**
$\quad\quad\quad t \leftarrow t_0 \frac{n}{N}$
$\quad\quad\quad \mathbf{z} \sim \mathcal{N}(\mathbf{0}, \mathbf{I})$
$\quad\quad\quad \epsilon \leftarrow \sqrt{\sigma^2(t) - \sigma^2(t - \Delta t)}$
$\quad\quad\quad \mathbf{x} \leftarrow \mathbf{x} + \epsilon^2 \mathbf{s}_{\boldsymbol{\theta}}(\mathbf{x}, t) + \epsilon \mathbf{z}$
$\quad\quad$ **end for**
$\quad$ **end for**
$\quad$ **Return** $\mathbf{x}$

---

---

**Algorithm 3** Guided image synthesis and editing with mask (VE-SDE)

---

**Require:** $\mathbf{x}^{(g)}$ (guide), $\boldsymbol{\Omega}$ (mask for edited regions), $t_0$ (SDE hyper-parameter), $N$ (total denoising steps), $K$ (total repeats)

$\quad \Delta t \leftarrow \frac{t_0}{N}$

$\quad \mathbf{x}_0 \leftarrow \mathbf{x}$

$\quad$**for** $k \leftarrow 1$ **to** $K$ **do**

$\quad\quad \mathbf{z} \sim \mathcal{N}(\mathbf{0}, \boldsymbol{I})$

$\quad\quad \mathbf{x} \leftarrow (\mathbf{1} - \boldsymbol{\Omega}) \odot \mathbf{x}_0 + \boldsymbol{\Omega} \odot \mathbf{x} + \sigma(t_0)\mathbf{z}$

$\quad\quad$**for** $n \leftarrow N$ **to** $1$ **do**

$\quad\quad\quad t \leftarrow t_0 \frac{n}{N}$

$\quad\quad\quad \mathbf{z} \sim \mathcal{N}(\mathbf{0}, \boldsymbol{I})$

$\quad\quad\quad \epsilon \leftarrow \sqrt{\sigma^2(t) - \sigma^2(t - \Delta t)}$

$\quad\quad\quad \mathbf{x} \leftarrow (\mathbf{1} - \boldsymbol{\Omega}) \odot (\mathbf{x}_0 + \sigma(t)\mathbf{z}) + \boldsymbol{\Omega} \odot (\mathbf{x} + \epsilon^2 \boldsymbol{s_\theta}(\mathbf{x}, t) + \epsilon\mathbf{z})$

$\quad\quad$**end for**

$\quad$**end for**

$\quad$**Return** x

---

---

**Algorithm 4** Guided image synthesis and editing (VP-SDE)

---

**Require:** $\mathbf{x}^{(g)}$ (guide), $t_0$ (SDE hyper-parameter), $N$ (total denoising steps), $K$ (total repeats)

$\quad \Delta t \leftarrow \frac{t_0}{N}$

$\quad \alpha(t_0) \leftarrow \prod_{n=1}^{N}(1 - \beta(\frac{nt_0}{N})\Delta t)$

$\quad$**for** $k \leftarrow 1$ **to** $K$ **do**

$\quad\quad \mathbf{z} \sim \mathcal{N}(\mathbf{0}, \boldsymbol{I})$

$\quad\quad \mathbf{x} \leftarrow \sqrt{\alpha(t_0)}\mathbf{x} + \sqrt{1 - \alpha(t_0)}\mathbf{z}$

$\quad\quad$**for** $n \leftarrow N$ **to** $1$ **do**

$\quad\quad\quad t \leftarrow t_0 \frac{n}{N}$

$\quad\quad\quad \mathbf{z} \sim \mathcal{N}(\mathbf{0}, \boldsymbol{I})$

$\quad\quad\quad \mathbf{x} \leftarrow \frac{1}{\sqrt{1-\beta(t)\Delta t}}(\mathbf{x} + \beta(t)\Delta t \boldsymbol{s_\theta}(\mathbf{x}, t)) + \sqrt{\beta(t)\Delta t}\, \mathbf{z}$

$\quad\quad$**end for**

$\quad$**end for**

$\quad$**Return** x

---

---

**Algorithm 5** Guided image synthesis and editing with mask (VP-SDE)

---

**Require:** $\mathbf{x}^{(g)}$ (guide), $\boldsymbol{\Omega}$ (mask for edited regions), $t_0$ (SDE hyper-parameter), $N$ (total denoising steps), $K$ (total repeats)

$\quad \Delta t \leftarrow \frac{t_0}{N}$

$\quad \mathbf{x}_0 \leftarrow \mathbf{x}$

$\quad \alpha(t_0) \leftarrow \prod_{i=1}^{N}(1 - \beta(\frac{it_0}{N})\Delta t)$

$\quad$**for** $k \leftarrow 1$ **to** $K$ **do**

$\quad\quad \mathbf{z} \sim \mathcal{N}(\mathbf{0}, \boldsymbol{I})$

$\quad\quad \mathbf{x} \leftarrow [(\mathbf{1} - \boldsymbol{\Omega}) \odot \sqrt{\alpha(t_0)}\mathbf{x}_0 + \boldsymbol{\Omega} \odot \sqrt{\alpha(t_0)}\mathbf{x} + \sqrt{1 - \alpha(t_0)}\mathbf{z}]$

$\quad\quad$**for** $n \leftarrow N$ **to** $1$ **do**

$\quad\quad\quad t \leftarrow t_0 \frac{n}{N}$

$\quad\quad\quad \mathbf{z} \sim \mathcal{N}(\mathbf{0}, \boldsymbol{I})$

$\quad\quad\quad \alpha(t) \leftarrow \prod_{i=1}^{n}(1 - \beta(\frac{it_0}{N})\Delta t)$

$\quad\quad\quad \mathbf{x} \leftarrow \Big\{(\mathbf{1} - \boldsymbol{\Omega}) \odot (\sqrt{\alpha(t)}\mathbf{x}_0 + \sqrt{1 - \alpha(t)}\mathbf{z}) + \boldsymbol{\Omega} \odot \Big[\frac{1}{\sqrt{1-\beta(t)\Delta t}}(\mathbf{x} + \beta(t)\Delta t \boldsymbol{s_\theta}(\mathbf{x}, t)) + $

$\quad \sqrt{\beta(t)\Delta t}\, \mathbf{z})\Big]\Big\}$

$\quad\quad$**end for**

$\quad$**end for**

$\quad$**Return** x

---

# D EXPERIMENTAL SETTINGS

## D.1 IMPLEMENTATION DETAILS

Below, we add additional implementation details for each application. We use publicly available pretrained SDE checkpoints provided by Song et al.; Ho et al.; Dhariwal & Nichol. Our code will be publicly available upon publication.

**Stroke-based image synthesis.** In this experiment, we use $K = 1, N = 500, t_0 = 0.5$, for SDEdit (VP). We find that $K = 1$ to 3 work reasonably well, with larger $K$ generating more realistic images but at a higher computational cost.

For StyleGAN2-ADA, in-domain GAN and e4e, we use the official implementation with default parameters to project each input image into the latent space, and subsequently use the obtained latent code to produce stroke-based image samples.

**Stroke-based image editing.** We use $K = 1$ in the experiment for SDEdit (VP). We use $t_0 = 0.5$, $N = 500$ for SDEdit (VP), and $t_0 = 0.45$, $N = 1000$ for SDEdit (VE).

**Image compositing.** We use CelebA-HQ (256×256) (Karras et al., 2017) for image compositing experiments. More specifically, given an image from CelebA-HQ, the user will copy pixel patches from other reference images, and also specify the pixels they want to perform modifications, which will be used as the mask in Algorithm 3. In general, the masks are simply the pixels the users have copied pixel patches to. We focus on editing hairstyles and adding glasses. We use an SDEdit model pretrained on FFHQ (Karras et al., 2019). We use $t_0 = 0.35$, $N = 700$, $K = 1$ for SDEdit (VE). We present more results in Appendix E.2.

## D.2 SYNTHESIZING STROKE PAINTING

**Human-stroke-simulation algorithm** We design a human-stroke-simulation algorithm in order to perform large scale quantitative analysis on stroke-based generation. Given a 256×256 image, we first apply a median filter with kernel size 23 to the image, then reduce the number of colors to 6 using the adaptive palette. We use this algorithm on the validation set of LSUN bedroom and LSUN church outdoor, and subset of randomly selected 6000 images in the CelebA (256×256) test set to produce the stroke painting inputs for Fig. 3a, Table 2 and Table 5. Additionally Fig. 30, Fig. 31 and Fig. 32 show examples of the ground truth images, synthetic stroke paintings, and the corresponding generated images by SDEdit. The simulated stroke paintings resemble the ones drawn by humans and SDEdit is able to generate high quality images based on this synthetic input, while the baselines fail to obtain comparable results.

**KID evaluation** KID is calculated between the real image from the validation set and the generated images using synthetic stroke paintings (based on the validation set), and the squared $L_2$ distance is calculated between the simulated stroke paintings and the generated images.

**Realism-faithfulness trade-off** To search for the sweet spot for realism-faithfulness trade-off as presented in Figure 3a, we select $0.01$ and every $0.1$ interval from $0.1$ to $1$ for $t_0$ and generate images for the LSUN church outdoor dataset. We apply the human-stroke-simulation algorithm on the original LSUN church outdoor validation set and generate one stroke painting per image to produce the same input stroke paintings for all choices of $t_0$. As shown in Figure 33, this algorithm is sufficient to simulate human stroke painting and we can also observe the realism-faithfulness trade-off given the same stroke input. KID is calculated between the real image from the validation set and the generated images, and the squared $L_2$ distance is calculated between the simulated stroke paintings and the generated images.

## D.3 TRAINING AND INFERENCE TIME

We use open source pretrained SDE models provided by Song et al.; Ho et al.; Dhariwal & Nichol. In general, VP and VE have comparable speeds, and can be slower than encoder-based GAN inversion

methods. For scribble-based generation on 256×256 images, SDEdit takes 29.1s to generate one image on one 2080Ti GPU. In comparison, StyleGAN2-ADA (Karras et al., 2020a) takes around 72.8s and In-domain GAN 2 (Zhu et al., 2020a) takes 5.2s using the same device and setting. We note that our speed is in general faster than optimization-based GAN inversions while slower than encoder-based GAN inversions. The speed of SDEdit could be improved by recent works on faster SDE sampling.

## E    EXTRA EXPERIMENTAL RESULTS

### E.1    EXTRA RESULTS ON LSUN DATASETS

**Stroke-based image generation.**    We present more SDEdit (VP) results on LSUN bedroom in Fig. 21. We use $t_0 = 0.5$, $N = 500$, and $K = 1$. We observe that, SDEdit is able to generate realistic images that share the same structure as the input paintings when *no paired data is provided*.

**Stroke-based image editing.**    We present more SDEdit (VP) results on LSUN bedroom in Fig. 22. SDEdit generates image edits that are both realistic and faithful to the user edit, while avoids making undesired modifications on pixels not specified by users. See Appendix D for experimental settings.

### E.2    EXTRA RESULTS ON FACE DATASETS

**Stroke-based image editing.**    We provide intermediate step visualizations for SDEdit in Fig. 23. We present extra SDEdit results on CelebA-HQ in Fig. 24. We also presents results on CelebA-HQ (1024×1024) in Fig. 29. SDEdit generates images that are both realistic and faithful (to the user edit), while avoids introducing undesired modifications on pixels not specified by users. We provide experiment settings in Appendix D.

**Image compositing.**    We focus on editing hair styles and adding glasses. We present more SDEdit (VE) results on CelebA-HQ (256×256) in Fig. 25, Fig. 26, and Fig. 27. We also presents results on CelebA-HQ (1024×1024) in Fig. 28. We observe that SDEdit can generate both faithful and realistic edited images. See Appendix D for experiment settings.

**Attribute classification with stroke-based generation.**    In order to further evaluate how the models convey user intents with high level user guide, we perform attribute classification on stroke-based generation for human faces. We use the human-stroke-simulation algorithm on a subset of randomly selected 6000 images from CelebA (256×256) test set to create the stroke inputs, and apply Microsoft Azure Face API[4] to detect fine-grained face attributes from the generated images. We choose gender and glasses to conduct binary classification, and hair color to perform multi-class classification on the images. Images where no face is detected will be counted as providing false and to the classification problems. Table 5 shows the classification accuracy, and SDEdit (VP) outperforms all other baselines in all attributes of choice.

### E.3    CLASS-CONDITIONAL GENERATION WITH STROKE PAINTING

In addition to user guide, SDEdit is able to also leverage other auxiliary information and models to obtain further control of the generation. Following Song et al. (2021) and Dhariwal & Nichol (2021), we present an **extra** experiment on class-conditional generation with SDEdit. Given a time-dependent classifier $p_t(\mathbf{y} \mid \mathbf{x})$, for SDEdit (VE) one can solve the reverse SDE:

$$\mathrm{d}\mathbf{x}(t) = \left[ -\frac{\mathrm{d}[\sigma^2(t)]}{\mathrm{d}t} (\nabla_{\mathbf{x}} \log p_t(\mathbf{x}) + \nabla_{\mathbf{x}} \log p_t(\mathbf{y} \mid \mathbf{x})) \right] \mathrm{d}t + \sqrt{\frac{\mathrm{d}[\sigma^2(t)]}{\mathrm{d}t}} \mathrm{d}\bar{\mathbf{w}} \qquad (13)$$

and use the same sampling procedure defined in Section 3.

---

[4]https://github.com/Azure-Samples/cognitive-services-quickstart-code/tree/master/python/Face

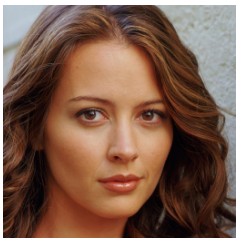 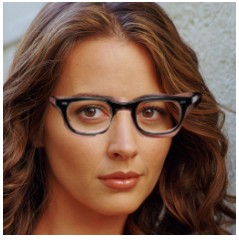 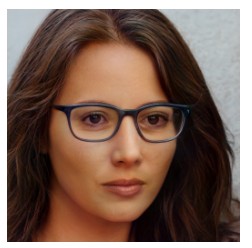 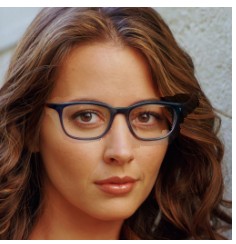

    (a) Dataset image         (b) User guide         (c) GAN output         (d) GAN blending

Figure 16: Post-processing samples from GANs by masking out undesired changes, yet the artifacts are strong at the boundaries even with blending.

| Methods | Gender | Glasses | Hair - Blond | Hair - Black | Hair - Grey |
|---|---|---|---|---|---|
| In-domain GAN 1 | 0.5232 | 0.6355 | 0.5527 | 0.5722 | 0.5398 |
| In-domain GAN 2 | 0.0202 | 0.0273 | 0.1806 | 0.3158 | 0.0253 |
| StyleGAN2-ADA | 0.0127 | 0.0153 | 0.1720 | 0.3105 | 0.0145 |
| e4e | 0.6175 | 0.6623 | 0.6731 | 0.6510 | 0.7233 |
| SDEdit (ours) | **0.8147** | **0.9232** | **0.8487** | **0.7490** | **0.8928** |

Table 5: Attribute classification results with simulated stroke inputs on CelebA. SDEdit (VP) outperforms all baseline methods in all attribute selected in the experiment. Details can be found in Appendix E.2.

For SDEdit (VP), we follow the class guidance setting in Dhariwal & Nichol (2021) and solve:

$$\mathbf{x}_{n-1} = \frac{1}{\sqrt{1-\beta(t_n)\Delta t}}(\mathbf{x}_n + \beta(t_n)\Delta t \mathbf{s}_{\boldsymbol{\theta}}(\mathbf{x}(t_n), t_n)) + \beta(t_n)\Delta t \nabla_{\mathbf{x}} \log p_t(\mathbf{y} \mid \mathbf{x}_n) + \sqrt{\beta(t_n)\Delta t} \, \mathbf{z}_n,$$

(14)

Fig. 34 shows the ImageNet (256×256) class-conditional generation results using SDEdit (VP). Given the same stroke inputs, SDEdit is capable of generating diverse results that are consistent with the input class labels.

### E.4 EXTRA DATASETS

We present additional stroke-based image synthesis results on LSUN cat and horse dataset for SDEdit (VP). Fig. 35 presents the image generation results based on input stroke paintings with various levels of details. We can observe that SDEdit produce images that are both realistic and faithful to the stroke input on both datasets. Notice that for coarser guide (*e.g.* the third row in Fig. 35), we choose to slightly sacrifice faithfulness in order to obtain more realistic images by selecting a larger $t_0 = 0.6$, while all the other images in Fig. 35 are generated with $t_0 = 0.5$.

### E.5 EXTRA RESULTS ON BASELINES

SDEdit preserves the un-masked regions automatically, while GANs do not. We tried post-processing samples from GANs by masking out undesired changes, yet the artifacts are strong at the boundaries. We further tried blending on GANs (GAN blending) with StyleGAN2-ADA, but the artifacts are still distinguishable (see Fig. 16).

## F HUMAN EVALUATION

### F.1 STROKE-BASED IMAGE GENERATION

Specifically, we synthesize a total of 400 bedroom images from stroke paintings for each method. To quantify sample quality, we ask the workers to perform a total of 1500 pairwise comparisons against SDEdit to determine which image sample looks more realistic. Each evaluation HIT contains 15 pairwise comparisons against SDEdit, and we perform 100 such evaluation tasks. The reward per task is kept as 0.2$. Since each task takes around 1 min, the wage is around 12$ per hour. For each question, the workers will be shown two images: one generated image from SDEdit and the other

**About this HIT:**

- **Please only participate in this HIT if you have normal color vision.**

- It should take about 1 minute.

- You will take part in an experiment involving visual perception. You'll see a series of pairs of images. In each pair, the images are "fake" images generated using a computer program. Your task is to determine which image is "more realistic".

Start!

Figure 17: The instruction shown to MTurk workers for pairwise comparison.

Which image do you think is **more realistic**?

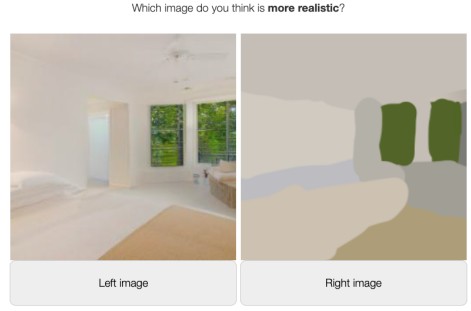

Left image          Right image

Figure 18: The UI shown to MTurk workers for pairwise comparison.

from the baseline model using the same input. The instruction is: "Which image do you think is **more realistic**" (see Fig. 17 and Fig. 18).

To quantify user satisfactory score (faithfulness+realism), we ask a different set of workers to perform another 3000 pairwise comparisons against SDEdit. For each question, the workers will be shown three images: the input stroke painting (guide), one generated image from SDEdit based on the stroke input, and the other from the baseline model using the same input. Each evaluation HIT contains 15 pairwise comparisons against SDEdit, and we perform 200 such evaluation tasks. The reward per task is kept as 0.2$. Since each task takes around 1 min, the wage is around 12$ per hour. The instruction is: "Given the input painting, how would you imagine this image to look like in reality? Choose the image that looks more reasonable to you. Your selection should based on how **realistic** and **less blurry** the image is, and whether it **shares similarities** with the input" (see Fig. 19 and Fig. 20).

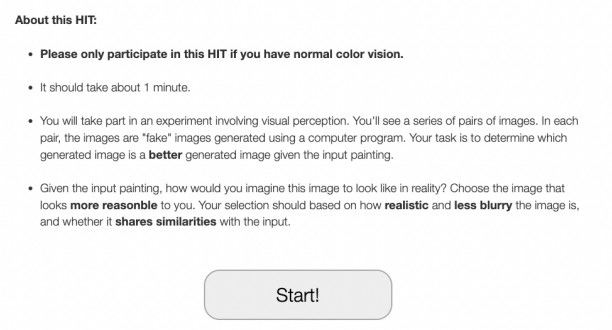

**About this HIT:**

- **Please only participate in this HIT if you have normal color vision.**

- It should take about 1 minute.

- You will take part in an experiment involving visual perception. You'll see a series of pairs of images. In each pair, the images are "fake" images generated using a computer program. Your task is to determine which generated image is a **better** generated image given the input painting.

- Given the input painting, how would you imagine this image to look like in reality? Choose the image that looks **more reasonble** to you. Your selection should based on how **realistic** and **less blurry** the image is, and whether it **shares similarities** with the input.

Start!

Figure 19: The instruction shown to MTurk workers for pairwise comparison.

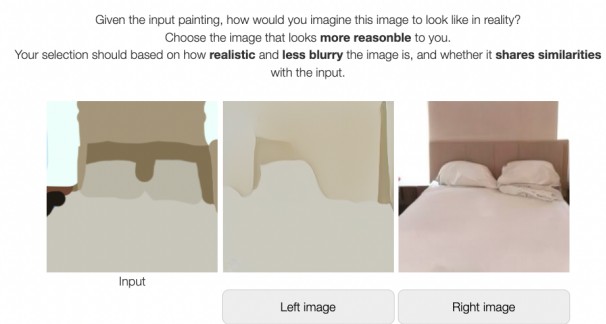

Figure 20: The UI shown to MTurk workers for pairwise comparison.

## F.2 IMAGE COMPOSITING ON CELEBA-HQ

To quantitatively evaluate our results, we generate 936 images based on the user inputs. To quantify realism, we ask MTurk workers to perform 1500 pairwise comparisons against SDEdit pre-trained on FFHQ (Karras et al., 2019) to determine which image sample looks more realistic. Each evaluation HIT contains 15 pairwise comparisons against SDEdit, and we perform 100 such evaluation tasks. The reward per task is kept as 0.2$. Since each task takes around 1 min, the wage is around 12$ per hour. For each question, the workers will be shown two images: one generated image from SDEdit and the other from the baseline model using the same input. The instruction is: "Which image do you think was **more realistic**?".

To quantify user satisfactory score (faithfulness + realism), we ask different workers to perform another 1500 pairwise comparisons against SDEdit pre-trained on FFHQ to decide which generated image matches the content of the inputs more faithfully. Each evaluation HIT contains 15 pairwise comparisons against SDEdit, and we perform 100 such evaluation tasks. The reward per task is kept as 0.2$. Since each task takes around 1 min, the wage is around 12$ per hour. For each question, the workers will be shown two images: one generated image from SDEdit and the other from the baseline model using the same input. The instruction is: "Which is a better polished image for the input? An ideal polished image should look **realistic**, and matches the input in visual appearance (e.g., they look like the same person, with matched hairstyles and similar glasses)".

Painting       Diverse outputs

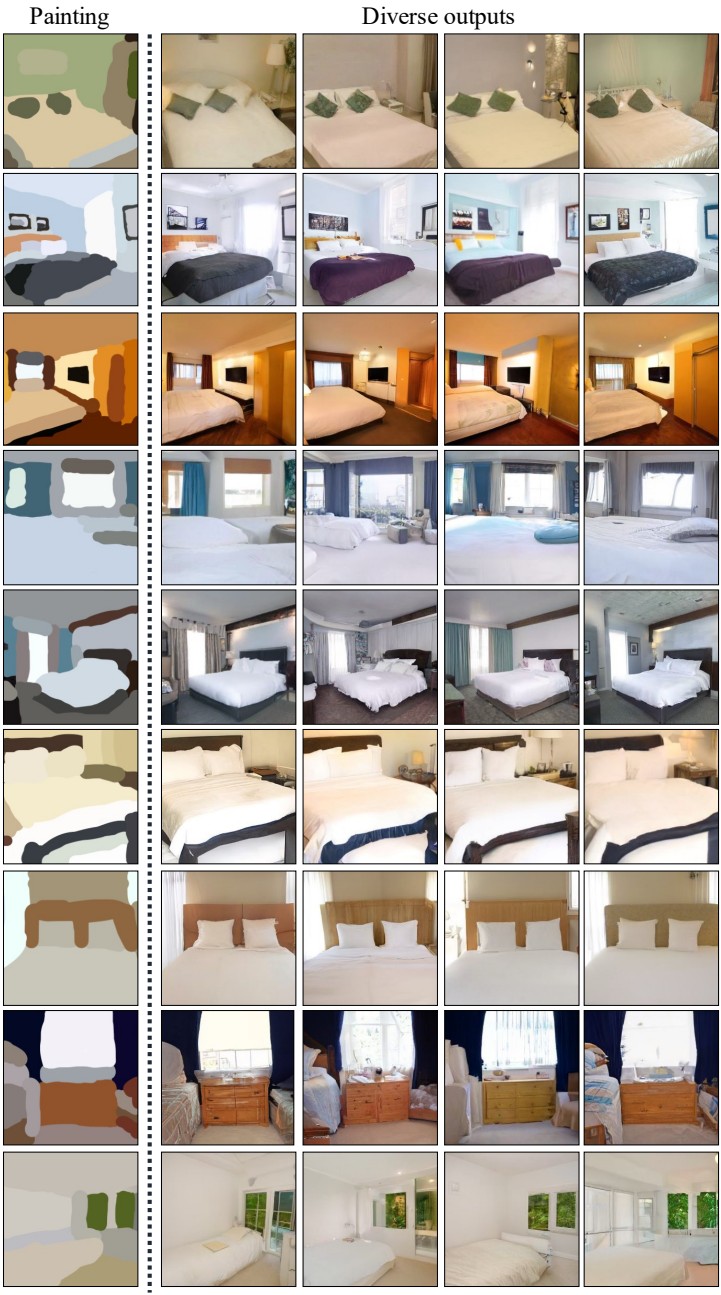

Figure 21: Stroke-based image generation on bedroom images with SDEdit (VP) pretrained on LSUN bedroom.

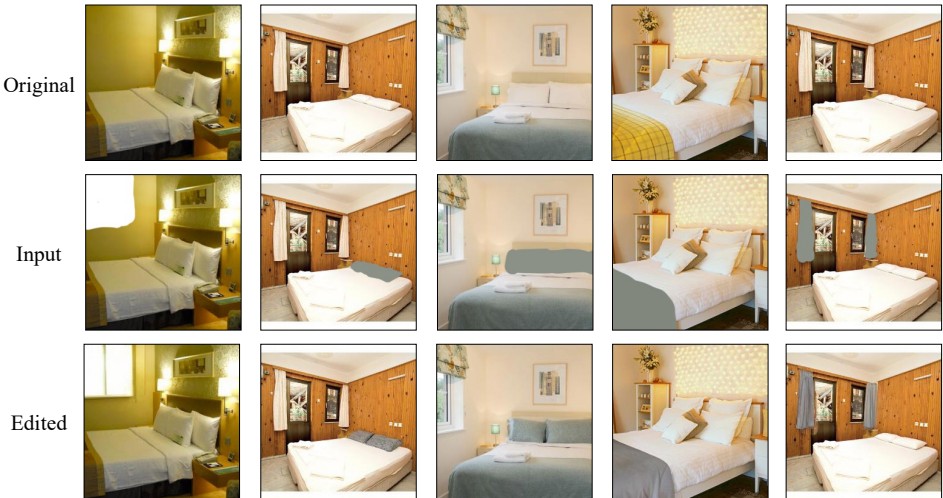

Figure 22: Stroke-based image editing on bedroom images with SDEdit (VP) pretrained on LSUN bedroom. SDEdit generates image edits that are both realistic and faithful (to the user edit), while avoids making undesired modifications on pixels not specified by users

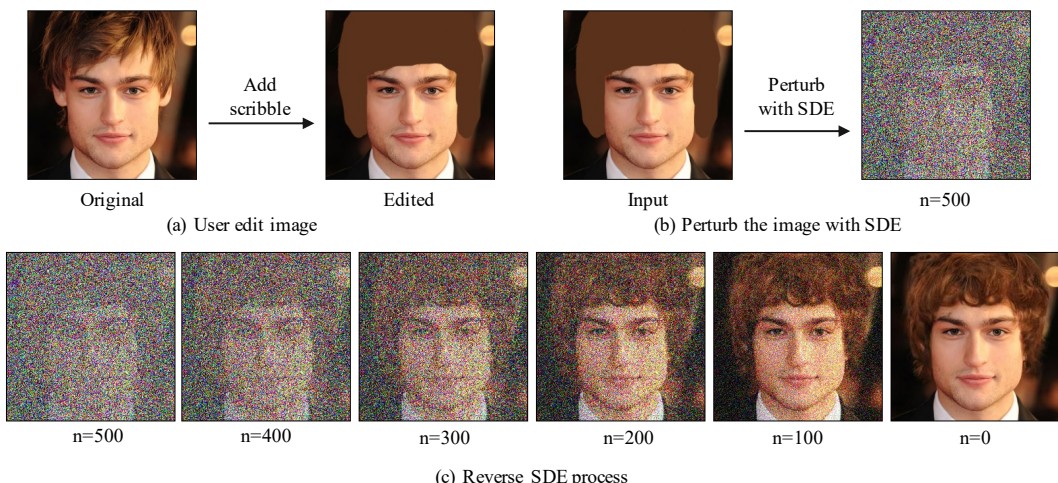

Figure 23: Stroke-based image editing. (a) Given an image, users will first modify the image using stroke, and provide a mask which describes the pixels covered by stroke. (b) The edited image will then be fed into SDEdit. SDEdit will first perturb the image with an SDE, and then simulate the reverse SDE (see Algorithm 5). (c) We provide visualization of the intermediate steps of reversing SDE used in SDEdit.

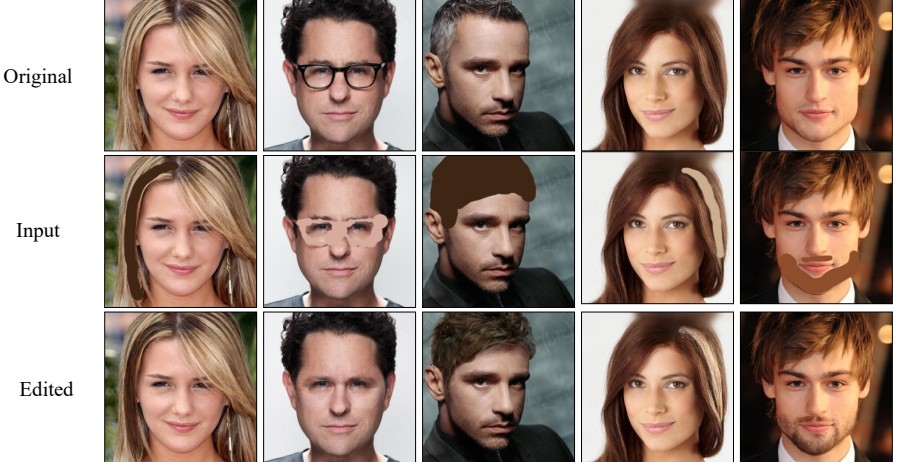

Figure 24: Stroke-based image editing on CelebA-HQ images with SDEdit. SDEdit generates image edits that are both realistic and faithful (to the user edit), while avoids making undesired modifications on pixels not specified by users.

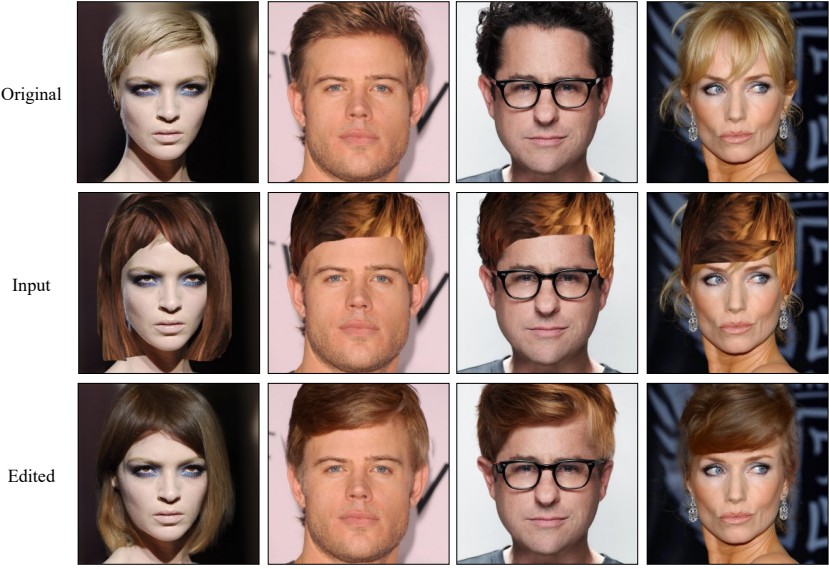

Figure 25: Image compositing on CelebA-HQ images with SDEdit. We edit the images to have brown hair. The model is pretrained on FFHQ.

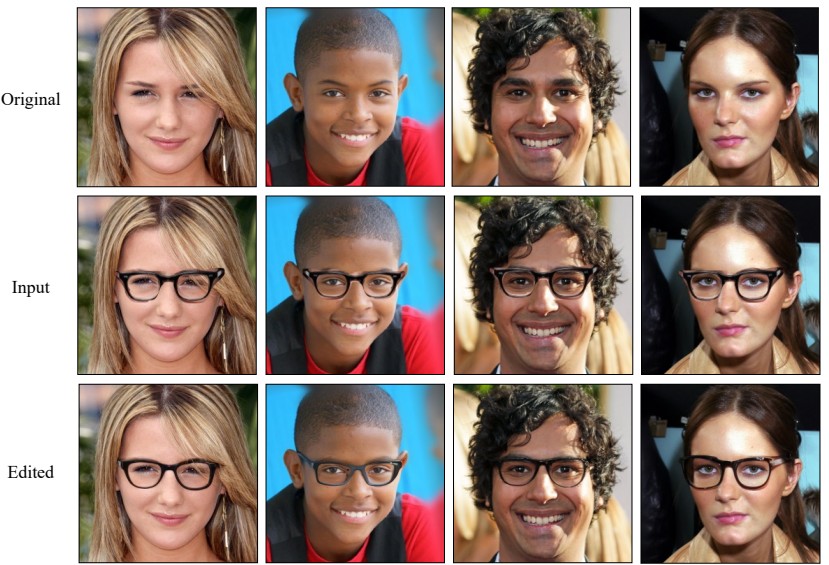

Figure 26: Image compositing on CelebA-HQ images with SDEdit. We edit the images to wear glasses. The model is pretrained on FFHQ.

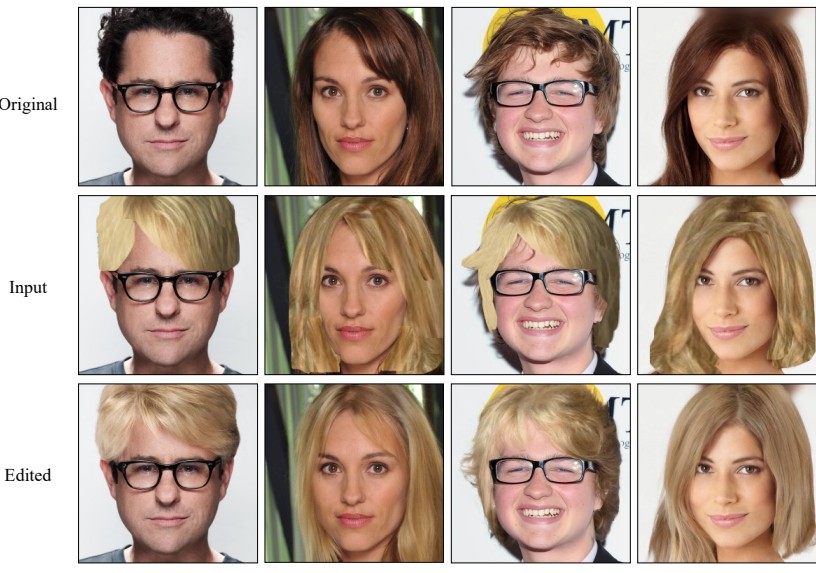

Figure 27: Image compositing on CelebA-HQ images with SDEdit. We edit the images to have blond hair. The model is pretrained on FFHQ.

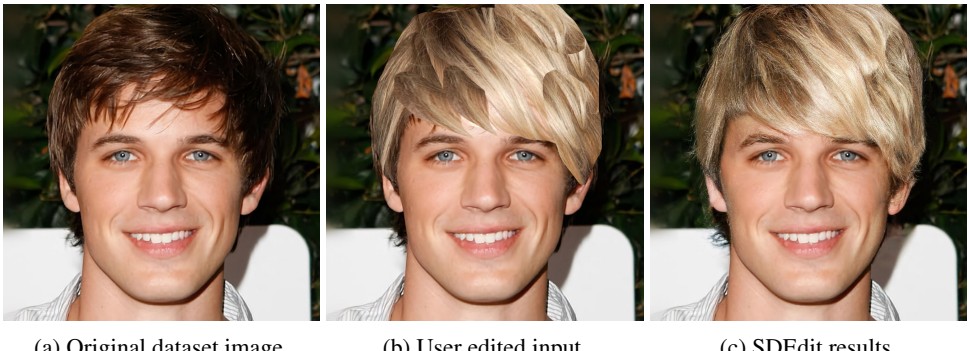

(a) Original dataset image.       (b) User edited input.       (c) SDEdit results.

Figure 28: Image compositing results with SDEdit (VE) on CelebA-HQ (resolution 1024×1024). The SDE model is pretrained on FFHQ.

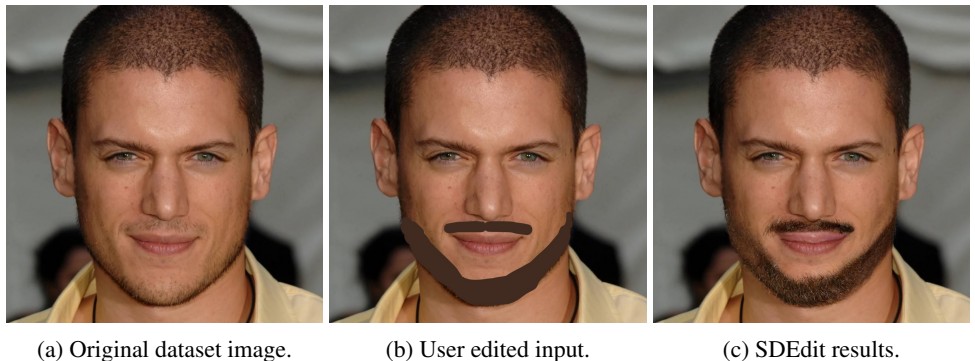

(a) Original dataset image.       (b) User edited input.       (c) SDEdit results.

Figure 29: Stroke-based image editing results with SDEdit (VE) on CelebA-HQ (resolution 1024×1024). The SDE model is pretrained on FFHQ.

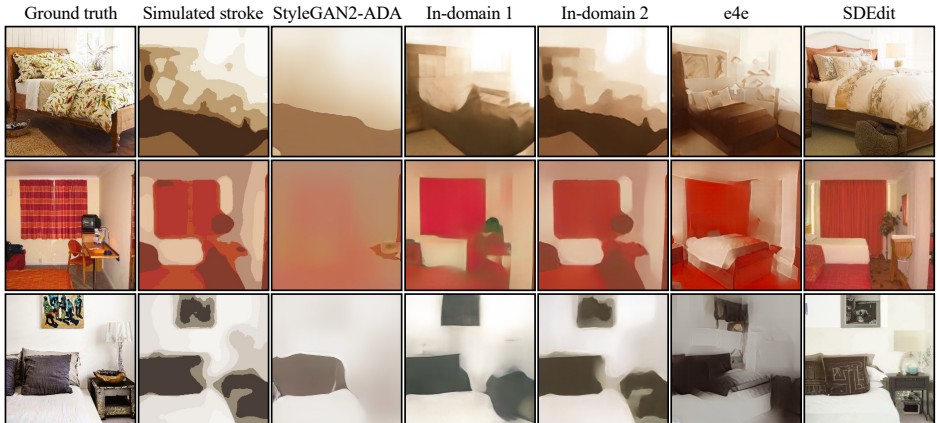

Figure 30: Stroke-based image generation with simulated stroke paintings inputs on bedroom images with SDEdit (VP) pretrained on LSUN bedroom dataset.

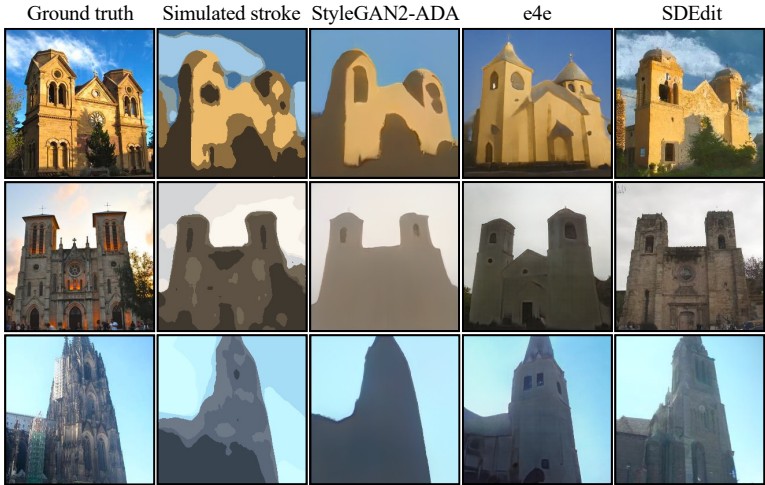

Figure 31: Stroke-based image generation with simulated stroke paintings inputs on church images with SDEdit (VP) pretrained on LSUN church outdoor dataset.

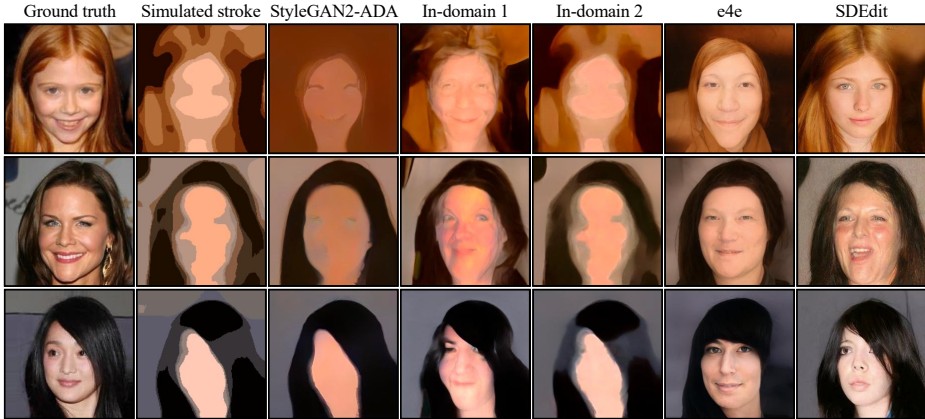

Figure 32: Stroke-based image generation with simulated stroke paintings inputs on human face images with SDEdit (VP) pretrained on CelebA dataset.

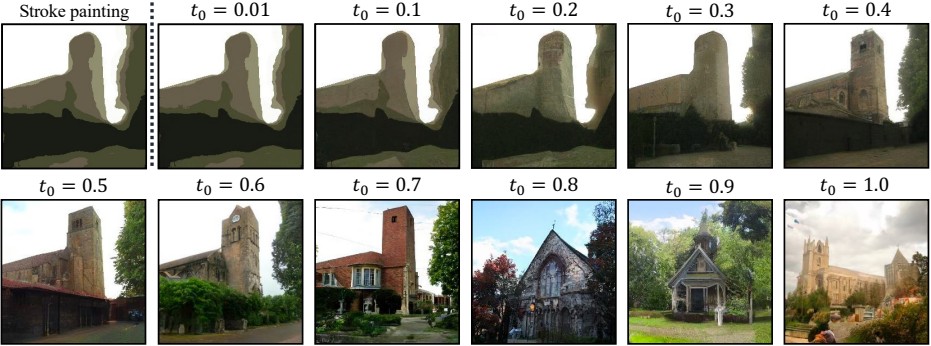

Figure 33: Trade-off between faithfulness and realism shown with stroke-based image generation with simulated stroke painting inputs on church images with SDEdit (VP) pretrained on LSUN church outdoor dataset.

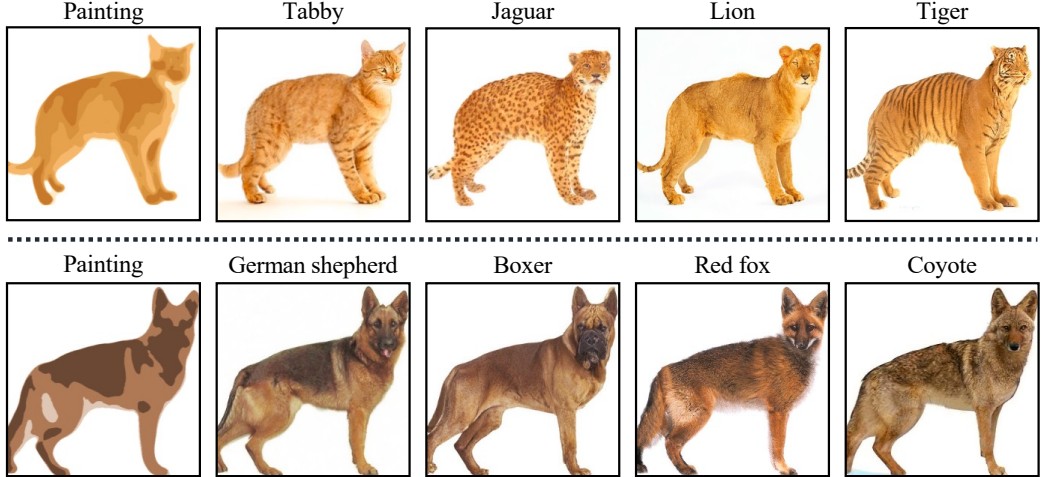

Figure 34: Class-conditional image generation from stroke paintings with different class labels by SDEdit (VP) pretrained on ImageNet.

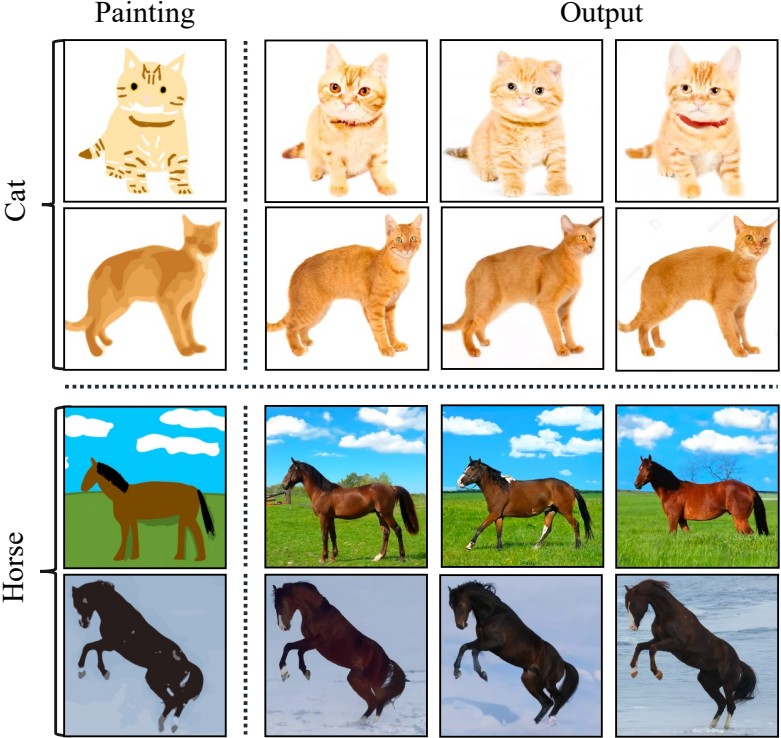

Figure 35: Stroke-based image generation with stroke inputs on cat and horse images with SDEdit (VP) pretrained on LSUN cat and horse dataset. Notice that for coarser guide (*e.g.* the third row), we choose to slightly sacrifice faithfulness in order to obtain more realistic images by selecting a larger $t_0 = 0.6$, while all the other images are generated with $t_0 = 0.5$.

