# OpenReview forum: "SDEdit: Guided Image Synthesis and Editing with Stochastic Differential Equations"
_ICLR.cc/2022/Conference — ICLR 2022 Poster_

### Official Review · Reviewer_Hvki · 2021-11-02

**Correctness:** 2
**Technical Novelty And Significance:** 2
**Empirical Novelty And Significance:** 2
**Recommendation:** 6
**Confidence:** 2

**Details Of Ethics Concerns:**

The paper is running experiment on human faces, this might raise concerns on the fairness and bias of human.

**Main Review:**

Strength:
1. The SDE formulation of image editing is an interesting approach, it's not only able to generate image with realism (through diffusion process) but also capture the real input (by controlling the time variable t0 that runs the diffusion process).

2. The proposed method is able to produce a diverse set of predictions given user input by sampling different Gaussian noise in the process.

3. The paper is well written and easy to follow.

Weakness

1. The paper is overclaimming. It is argued that the proposed method is able to diffuse the image "given a user guide of any type", however, it seems the proposed method is only able to handle the user guide in a form of manipulating pixel rgb values (strokes, color patch, or image patch), there are many other user guides exist, such as semantic mask (GauGAN), or text (using CLIP model), and it's not clear how the proposed method generalize to these user guides.

2. The proposed method need to run iteratively to obtain the final image, which makes the inference time inevitably long. It would be great to include the running time of the proposed method in comparing with other methods.

3. The main technique in the paper (The SDE process) is previous method (Song et. al. 2020, 2021), the paper only proposed to use the human edited image as initialization of SDE at a particular time t0, thus the novelty is not great enough.

**Summary Of The Paper:**

The paper proposed Stochastic Differentiable Image Editing method which iteratively diffuse the human edited image  to real images, using the pretrained score-based SDE models (the edit is in a form of manipulating image pixel values, e.g. stroke, color paint, or image patch).The method achieves both realistic image editing and faithfulness to human input through user studies.

**Summary Of The Review:**

The paper proposed an interesting technique to edit image using the SDE. The generated image is not only faithful but also relistic. However, considering the technique novelty and the overclaiming, I vote for a weak reject initially and am willing to listen to the authors and other reviewers.

---

> ### Author Response · Authors · 2021-11-23
> **Incorporated feedback in the revision.**
>
> We thank you for the constructive feedback!
>
> **Q: "Given a user guide of any type" is being overclaiming**.
>
> **A:** We thank you for pointing it out! We meant to refer to "user guide in a form of manipulating pixel rgb values”, just as you have pointed out.  We agree that our previous description might not be precise. Based on your feedback, we have changed it to "user guide in a form of manipulating RGB pixels” in the revision.
>
> **Q: Running time of SDEdit.**
>
> **A:** We reported inference time of SDEdit in Appendix C.3 in the original submission (now Appendix D.3). SDEdit takes around 29.1s to generate one 256*256 image on one 2080Ti GPU. In comparison, StyleGAN2-ADA takes around 72.8s and In-domain GAN 2 takes 5.2s using the same device and setting. We note that our speed is in general faster than optimization-based GAN inversions, while slower than encoder-based GAN inversions. Further, while SDEdit is slower than some GAN-based methods, it produces much more realistic samples than GANs do as shown in Table 1.
>
> We believe recent works [1,2] in speeding up the sampling of score-based / diffusion-based models, which have achieved a more than 100 times speedup, can be applied to improve the sampling speed of SDEdit. We leave the incorporation of such faster sampling approaches into SDEdit as future work.
>
> [1] https://openreview.net/forum?id=TIdIXIpzhoI
>
> [2] https://arxiv.org/abs/2106.00132
>
> **Q: Technical novelty.**
>
> **A:** We respectfully disagree. Methods proposed by Song et. al. introduce an extra noise-conditioned classifier for conditional generation and the performance of the classifier is critical to the conditional generation performance. Their settings are more similar to regular inverse problems where the measurement function is known (we discussed this in Section 3 of the original submission). Since we do not have a known “measurement” function for user-generated guides, their approach cannot be directly applied to user-guided image synthesis or editing in the form of manipulating pixel RGB values. To deal with this limitation, SDEdit initializes the reverse SDE based on user input and modifies $t_0$ accordingly---an approach different from Song et. al. (which always use the same initialization). This technique allows SDEdit to achieve faithful and realistic image editing or generation results without extra task-specific model learning (e.g., an additional classifier in Song et. al.). We further analyze the realism-faithfulness trade-off with Proposition 1, which is also not proposed by previous works.
>
> SDEdit achieves state-of-the-art performance on challenging user-guided tasks such as stroke-based image synthesis and editing as well as image compositing, where a direct application of Song et. al. 2020, 2021 will fail by generating random inpainting results that are not faithful to the user input (see Appendix B.5, Figure 15).
>
> **Q: Ethical concerns: "The paper is running experiments on human faces”**
>
> **A:**  Just like all image synthesis and editing algorithms, our method has both positive and negative societal impacts depending on the applications and usages. On the positive side, SDEdit enables everyday users with or without artistic expertise to create and edit photo-realistic images with minimum effort, lowering the barrier to entry for visual content creation. On the other hand, as other deep-learning models (such as GAN-based methods for face-editing), SDEdit might be exploited by malicious users. In our code release, we will explicitly specify allowable uses of our system with appropriate licenses. We have included this discussion in the revision.

---

### Official Review · Reviewer_CnCY · 2021-11-02

**Correctness:** 4
**Technical Novelty And Significance:** 2
**Empirical Novelty And Significance:** 3
**Recommendation:** 8
**Confidence:** 4

**Main Review:**

Strengths:
The whole idea is interesting, effective and simple. The SDEdit perturbs the input image with the forward SDE and generates edits by reversing the SDE, the SDE is pretrained on unlabeled data, so the SDEdit does not require collecting pair images. Meanwhile, since the excellent generation performance of SDE, the SDEdit can generate the content with high quality. The mask strategy can help the user edit the regions where they want.

Weaknesses:
The SDEdit can not edit the image flexibly enough, such as closing eyes, opening mouth, changing lip color. Meanwhile, the guidance can not influence the results well sometimes (Fig.1 Stroke-based Editing), there is no brown forest in the region where the user draws the brown stroke. Finally, the Stroke-based Editing is not complex and some methods can do this such as [1][2], the SDEdit should compare with them.

[1]DeFLOCNet: Deep Image Editing via Flexible Low-level Controls. CVPR2021

[2]Sc-fegan: Face editing generative adversarial network with user’s sketch and color. ICCV 2019

**Summary Of The Paper:**

The paper proposesed an image editing and synthesis system (SDEdit) with SDE. By injecting the guided image into the reverse process with appropriate $t_0$, the results can have a trade-off between realism and faithfulness.

**Summary Of The Review:**

The paper is well-written and easy to follow. The results are impressive. Although Stroke-based Editing is not complex, especially on the face,  I think this is a good start of using the SDE to do the image editing task.

---

> ### Author Response · Authors · 2021-11-23
> **Extra results and ablation studies of SDEdit are added.**
>
> We thank you for the constructive feedback!
>
> **Q:  Flexible image editing (such as closing eyes, opening mouth, changing lip color) with SDEdit.**
>
> **A:**  As requested, we have performed extra image editing experiments including editing closing eyes, opening mouth, and changing lip color. We included the results in Appendix B. We observe that SDEdit can still achieve successful editing results (see Appendix B.2), which provides additional evidence that SDEdit can also be used for certain flexible image editing tasks.
>
> **Q: "The guidance can not influence the results well sometimes (Fig.1 Stroke-based Editing), there is no brown forest in the region where the user draws the brown stroke.”**
>
> **A:** In our experiment, we use a shared $t_0$ for all guides in the same task, and we empirically observe that the shared $t_0$ works well for all reasonable guides in the same task (see **Section 3**, **Choice of $t_0$**). As illustrated in Figure 3, we can further tune $t_0$ to tradeoff between faithfulness and realism for individual images---with a smaller $t_0$ corresponding to a more faithful but (potentially) less realistic generated image. If we want to keep the brown stroke, we can reduce $t_0$ to increase its faithfulness while slightly decreasing its realism. To show this, we include extra analysis using different $ t_0$ for Fig. 1 in Appendix B.3.
>
> **Q: "The Stroke-based Editing is not complex and some methods can do this such as [1][2], the SDEdit should compare with them.”**
>
> **A:** We thank you for pointing out references [1, 2]. Unlike SDEdit, both methods [1, 2] require input-output pairs  (e.g., sketch images or color images) during training, which requires paired data collection (or generation) and model re-training for new editing tasks. Unlike previous work [1, 2], SDEdit does not require paired data (e.g., sketch or color images) during training and a single model can be directly applied to various guided image synthesis and editing tasks using a pre-trained SDE-based generative model without task-specific training (see Fig.1).
>
> In the rebuttal, we have performed additional comparisons with SC-FEGAN [2] on stroke-based image synthesis and editing, as shown in Appendix B.4. We observe that SDEdit performs better than SC-FEGAN [2] on stroke-based synthesis and editing tasks in terms of realism. We were hoping to compare with DeFLOCNet [1], but the code is not available in their official repo. We are happy to include a comparison with [1], if the code becomes available in the future. We have included discussions of both work [1,2] in related work.
>
> [1] DeFLOCNet: Deep Image Editing via Flexible Low-level Controls. CVPR 2021
>
> [2] SC-FEGAN: Face editing generative adversarial network with user’s sketch and color. ICCV 2019

---

> > ### Comment · Reviewer_CnCY · 2021-11-29
> > **Thanks for the author's response.**
> >
> > Thanks for the author's response. The rebuttals have addressed my concern, and I would like to increase my score.

---

### Official Review · Reviewer_N9v6 · 2021-11-03

**Correctness:** 3
**Technical Novelty And Significance:** 3
**Empirical Novelty And Significance:** 2
**Recommendation:** 6
**Confidence:** 3

**Main Review:**

Stroke-based image editing algorithms have been widely-adopted and popular for many artistic design tools and softwares. Many of those are (conditional) GAN-based methods, and can be often challenging to balance the tradeoff between use control vs. result faithfulness.

The proposed method has a novel technical contribution, that utlizes iterative SDE denoising for guided stroke sketch input + perturbed noise, with photorealistic output via the reverse of the SDE. The paper has compared with a few state-of-the-art baselines and demosntrate its effectiveness in producing more faithful editing results.

One question I have is that, how good the guided stroke sketch needs to be? I don't see ablation study on this, but I think it is an important factor for practical applications.

Paper writing and references seem fine to me.

**Summary Of The Paper:**

This paper proposes to synthetize and edit realistic images based on stocahstic differential equations (SDE). The method is based on diffusion generative model that iteratively denoises through SDEs and demosntrates that this SDE prior increases photorealism of the images, compared to state-of-the-art GAN-based approaches.

**Summary Of The Review:**

Overall I am leaning towards a weak accept. The paper has clear novelty regarding stroke-based image editing via perturb + reverse SDEs. Comparisons and ablation study look good to me except for the one on the level of user sketch control.

---

> ### Author Response · Authors · 2021-11-23
> **Extra analysis on how good the guided stroke needs to be are provide in the revision**
>
> We thank you for the constructive feedback!
>
> **Q: "How good the guided stroke sketch needs to be?”**
>
> **A:** As discussed in the *Realism-faithfulness trade-off* section (Section 3 in the original submission), if the guide is far from any realistic image (e.g., random noise or sketch has an unreasonable composition), then we must tolerate at least a certain level of deviation from the guide (non-faithfulness) in order to produce a realistic image. This idea is further illustrated in Figure 3 and Proposition 1.
>
> For practical applications, we have performed extra ablation studies on how the quality of the guided stroke affects the results and included the results in Appendix B.1. Specifically, we consider stroke input of
> 1) a human face with limited detail for a CelebA-HQ model,
> 2) a human face with spikes for a CelebA-HQ model,
> 3) a building with limited detail for a LSUN-church model,
> 4) a horse for a LSUN-church model.
>
> We observe that SDEdit is in general tolerant to different kinds of user inputs.
>
> We also quantitatively analyze the effect of user guide quality using simulated stroke paintings as input. Described in Appendix D.2, the human-stroke-simulation algorithm uses different numbers of colors to generate stroke guides with different levels of detail. We compare SDEdit with baselines qualitatively in Fig.9 and quantitatively in the following table (Table 4 in the paper). Similarly, we observe that SDEdit has a high tolerance to input guides and consistently outperforms the baselines across all setups in this experiment.
>
> | # of colors 	| StyleGAN2-ADA 	|       	|   e4e  	|       	| SDEdit (Ours) 	|       	|
> |:-----------:	|:-------------:	|:-----:	|:------:	|:-----:	|:-------------:	|:-----:	|
> |             	|      KID     $\downarrow$ 	|   $L_2$  $\downarrow$	|   KID $\downarrow$ 	|   $L_2$  $\downarrow$	|      KID     $\downarrow$ 	|   $L_2$  $\downarrow$	|
> |      3      	|     0.1588    	| 67.22 	| 0.0379 	| 70.73 	|     **0.0233**    	|  **36.00** 	|
> |      6      	|     0.1544    	| 72.41 	| 0.0354 	| 68.53 	|      **0.0156**    	|  **37.67** 	|
> |      16     	|     0.0923    	| 69.52 	| 0.0319 	| 68.20 	|      **0.0135**   	|  **37.70** 	|
> |      30     	|     0.0911    	| 67.11 	| 0.0304 	| 68.66 	|      **0.0128**   	|  **37.42** 	|
> |      50     	|     0.0922    	| 65.28 	| 0.0307 	| 68.80 	|      **0.0126**    	|  **37.40** 	|

---

### Author Response · Authors · 2021-11-23
**We thank all reviewers for the constructive feedback**

We thank all reviewers for the constructive feedback. We are glad that all reviewers acknowledge the effectiveness of our approach. In the revision, we have included a new section for extra ablation studies (Appendix B). In the following, we answer reviewers’ questions one by one. Specifically, we provide extra ablation studies on how good the guided stroke needs to be (*Reviewer N9v6*); we provide extra analysis on flexible image editing and extra comparison with other baselines (*Reviewer CnCY*); we revise the writing, clarify the contribution, compare running time, and include extra discussion in the Ethics Statement section (*Reviewer Hvki*). We use blue text for changes in the revision.

---

### Decision · Program_Chairs · 2022-01-20

**Decision:**

Accept (Poster)

**Comment:**

Thank you for your submission to ICLR.

This paper presents a technique for image synthesis based on stochastic differential equations and a diffusion model.  This looks to be a very nice idea with good results.  After discussion, the reviewers converged and all agreed that the paper is ready for publication---the most negative reviewer raised their score after the author rebuttal, from a weak reject to weak accept.  The rebuttal clearly and concisely addressed several concerns of the reviewers.

I'm happy to recommend accepting the paper.